# Affiliative behaviours regulate allostasis development and shape biobehavioural trajectories in horses

Mathilde Valenchon [1,6] ✉, Fabrice Reigner[2], Gaëlle Lefort [1], Hans Adriaensen[1,3], Amandine Gesbert[2], Philippe Barrière[2], Yvan Gaude[2], Frederic Elleboudt[1,3], Isabelle Lévy[4], Camille Ducluzeau[4], Joëlle Dupont[1], Anne-Lyse Lainé[1], Ivy Uszynski [5], Hugues Dardente[1], Cyril Poupon[5], Léa Lansade[1], Ludovic Calandreau[1], Matthieu Keller [1] ✉ & David André Barrière [1] ✉

Social interactions shape both the physiological and behavioural development of offspring, and poor care/early caregiver loss is known to promote adverse outcomes during infancy in both animals and humans. How affiliative behaviours impact the future development of offspring remains an open question. Here, we used *Equus caballus* (domestic horse) as a model to investigate this question. By coupling magnetic resonance imaging, longitudinal biobehavioural assessments and advanced multivariate statistical modelling, we found that prolonged maternal presence during infancy promotes the maturation of brain regions involved in both social behaviour (anterior cingulate cortex and retrosplenial cortex) and physiological regulation (hypothalamus and amygdala). Additionally, offspring benefiting from a prolonged maternal presence showed higher default mode network connectivity, improved social competences and feeding behaviours, and higher concentrations of circulating lipids (triglyceride and cholesterol). The findings of the present study underscore the salient role of social interactions in the development of allostatic regulation in offspring.

In large social mammals, including humans, caregivers—typically parents—provide care during a species-specific immature period beyond the lactation period to ensure the acquisition of the social skills necessary for the survival and reproductive fitness of the offspring[1–3]. Caregiver death/loss or poor caregiving during this socially immature period is known to promote adverse physiological and behavioural outcomes in adulthood in both animals and humans[2,4]. In wild primates considered altricial (e.g. mountain gorillas), longitudinal studies have shown that offspring are more likely to die early in adulthood when mothers experience poor physical conditions and display poorer abilities to raise their progeny[1,2]. In precocial species such as horses, previous studies have shown that an abrupt rupture of the maternal bond at approximately 4–6 months of age induces long-lasting emotional/social outcomes, oral/locomotor stereotypies and physiological impairments in cortisol/lipid metabolism associated with a reduction in telomer length in offspring[5–7]. Hence, regardless of brain maturity at

[1]CNRS, INRAE, Université de Tours, UMR Physiologie de la Reproduction et des Comportements, Nouzilly, France. [2]INRAE, Unité Expérimentale de Physiologie Animale de l'Orfrasière, Centre de Recherche de Tours, Nouzilly, France. [3]INRAE, Université de Tours, CHU de Tours, PIXANIM, Nouzilly, France. [4]Veterinary Clinic of the Nouvetière, La Nouvetière, Sonzay, France. [5]BAOBAB, NeuroSpin, Université Paris-Saclay, CNRS, CEA, Gif-sur-Yvette, France. [6]Present address: Université Paris-Saclay, INRAE, AgroParisTech, UMR Modélisation Systémique Appliquée aux Ruminants, Palaiseau, France. ✉ e-mail: mathilde.valenchon@inrae.fr; matthieu.keller@inrae.fr; david.barriere@cnrs.fr

birth, it seems that in large social mammals, affiliative behaviours provide social interactions salient enough to foster the development of regulatory systems vital for the control of physiology and behaviours in adulthood[3,4,8].

During the past decade, allostasis, a brain-centred concept, has been proposed to explain this phenomenon[4,9–13]. Allostasis posits that the brain is dedicated to predictive and anticipatory regulation of the internal milieu through the continuous integration of internal/external cues necessary for survival[9,14,15]. In the adult human brain, allostasis would be supported by a complex system that integrates the cortex (cingulate, insula and prefrontal cortex), hippocampus, striatum, hypothalamus, amygdala, periaqueductal grey (PAG) and parabrachial nucleus[16–19]. In this system, cortical areas are embedded within domain-general core networks such as the 'default mode network' (DMN), in which intero/exteroceptive inputs are constantly integrated to construct predictive models. Primary sensory cortices would then compare these models, and the difference (i.e. error prediction) would be used as the final output to tune the autonomic system. This allostatic regulatory system would control the activity of peripheral organs (e.g. adrenals) through the release of endocrine hormones by the hypothalamus and brainstem regions (e.g. PAG), regulating homoeostasis (e.g. heart rate) and behaviours (e.g. fight/flight). Notably, the level of error prediction, encoded by striatal dopamine levels, would support learning mechanisms used to refine forthcoming predictions and minimise energy requirements for future adaptations[4,20]. Therefore, the allostatic regulatory system is an acquired system that develops during infancy and is further refined throughout life through physical and social experiences. This system is key for the daily physiological/behavioural adaptations of an individual to navigate in his or her environment[3,4,8].

To date, the literature supports the hypothesis that the initial caregiver–offspring dyad is a critical scheme in which affiliative behaviours, fuelled by the maternal bond, provide the social cues necessary for the maturation of the allostatic regulatory system during the socially immature period[1–3]. Large social mammals would share this process, that is vital for the adaptation of offspring to a challenging physical/social environment. However, we still do not know how social interactions engage these brain-maturing processes, what these processes and their neural substrates are, or how they eventually shape physiological and behavioural trajectories during development. In this work, we combine in vivo MRI with longitudinal behavioural and physiological assessments in domestic horses to examine how prolonged maternal presence during mid-infancy influences brain and biobehavioural development. We show that prolonged maternal presence is associated with the maturation of allostasis-related brain regions, increased DMN-like connectivity, and linked changes in social behaviour, time spent feeding, growth and lipid metabolism.

## Results

### Use of *Equus caballus* to study brain development and biobehavioural trajectories

We chose to evaluate how affiliative behaviours during infancy influence the development of the allostasis-regulating system and further shape the biobehavioural trajectories of offspring in *Equus caballus* (domestic horse). The mother–offspring dyad ontogeny in horses shares many features with that of large social mammalian species such as ungulates, cetaceans and primates, including humans, which makes it an excellent model for such investigations. Under free-ranging conditions, mares typically have one to two offspring per gestation and provide an extended period of parental care (>1 year), and mothers rapidly establish a strong, long-lasting and individualised affiliative bond with their offspring[21]. Under human care, however, weaning traditionally occurs at approximately 6 months of age and results from human-driven separation of the mother and foal rather than a spontaneous process. We developed a

holistic approach based on in vivo magnetic resonance imaging (MRI) to study brain structure and function in horse offspring associated with a longitudinal collection of behavioural and physiological data recorded between 6 and 13 months of age in two groups of animals raised either in naturalistic or human-driven conditions to test our hypothesis (Fig. 1a). In horse ontogeny, this timeframe is considered to include a childhood-like period starting when offspring can feed themselves and control their physiology (approximately 6 months) but are still socially dependent on their mothers and terminating with the onset of sexual maturation, which marks the beginning of adolescence (approximately 13 months)[22]. During this timeframe, we applied our multimodal approach to two animal groups (a total of 24 animals) herded under the same conditions (indoor collective stalls with free access to an outdoor paddock). The first group, named '*with mother*', was composed of unweaned offspring (6 males/6 females), which had been raised with their mothers (12 mares). The second group, named '*without mother*', was composed of 12 weaned offspring (6 males/6 females), which had been separated from their mothers at the beginning of the experiment at 6 months of age (Fig. 1b). In total, 36 animals (offspring plus mothers) were herded together. This procedure allows us to study the effects of caregiver loss on brain development and biobehavioural trajectories until adolescence and minimise biases induced by not only sex hormones but also stress and/or emotional contagion through the promotion of a socially enriched environment for weaned animals. Multimodal MRI data (functional, diffusion and anatomical) were collected 1 month after the weaning procedure in both groups to evaluate both functional and microstructural changes between groups and to explore the effects on brain development in foals (Fig. 1c). In vivo imaging data were used to create the in vivo brain template and atlas of *Equidae*, the Turone Equine Brain Template and Atlas, which was used to analyse the MRI data (see Supplementary Fig. 2 and Supplementary Table 1). The role of extended maternal care on foals' behavioural development was explored by performing multiple evaluations at 1, 3 and 7 months after the weaning procedure to assess the activity budgets, spatial proximity, centrality and spontaneous behavioural activity of each animal. Additionally, sociability and fearfulness were evaluated by conducting standardised behavioural tests 7 months after weaning (Fig. 1d). Finally, as a method to determine the effect of prolonged maternal presence on the physiological development of the offspring, the animals were weighed, and blood was sampled at 1, 3 and 7 months to assess the plasma levels of basal cortisol, oxytocin, glycaemia, cholesterol, IGF-1 and insulin (Fig. 1e).

### A multimodal approach discriminates the effect of prolonged maternal presence

At the end of the experiment, a total of 76 variables (see Supplementary Data 1) were collected for 23 animals, as one female belonging to the '*with mother*' group was removed from the experiment for medical reasons. From this dataset, we first conducted a holistic exploratory analysis to determine the potential for discriminating individuals based on treatment ('*with mother*' versus '*without mother*'). We performed multiple factor analysis (MFA) on all the parameters for which the variables were grouped into equally weighted sets for analysis to achieve this goal, and six groups of variables were defined: *weight gain, non-social behaviour, non-social personality traits, sociality, MRI* and *physiology*. MFA showed clear separation between the '*with mother*' (orange) and '*without mother*' (purple) groups across the first dimension, demonstrating the multidimensional impact of extended maternal care on offspring development (Fig. 1f). Physiological variables exhibited the most significant disparities between groups, followed by behavioural variables and, finally, specific brain metrics (see Supplementary Data 1) from the brain regions involved in the allostasis regulatory system (Fig. 1g).

## a. Calendar

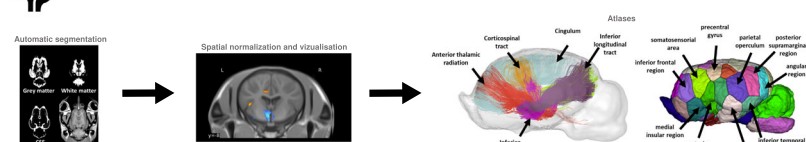

## b. Experimental design

**With** *vs.* **Without**
mother beyond the age of 6 months

## c. Brain development

**MRI acquisition** (sequence MRI, diffusion MRI and rs-fMRI)

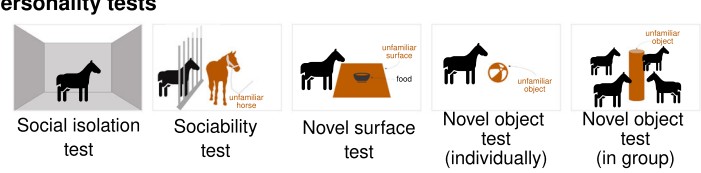

*Creation of the Turone Equine Brain Template & Atlas Toolkit*
*for MRI data analysis and vizualisation in horses*

## d. Behavioural development

**Direct and video analyses** Day time (10am-3:30pm)

**Scan sampling / 15min** proximity with closest neighbour + Behavioral activity
**SNA + activity budgets**

**Personality tests**

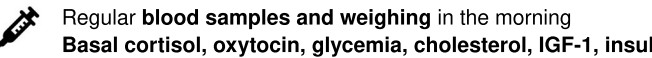

Social isolation test | Sociability test | Novel surface test | Novel object test (individually) | Novel object test (in group)

## e. Physiological development

Regular **blood samples and weighing** in the morning
**Basal cortisol, oxytocin, glycemia, cholesterol, IGF-1, insulin**

## f. Exploratory analysis

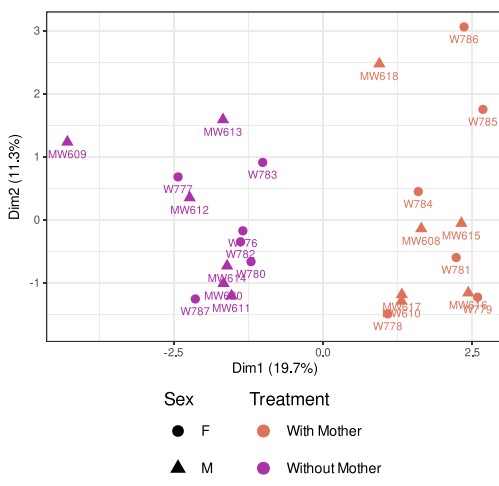

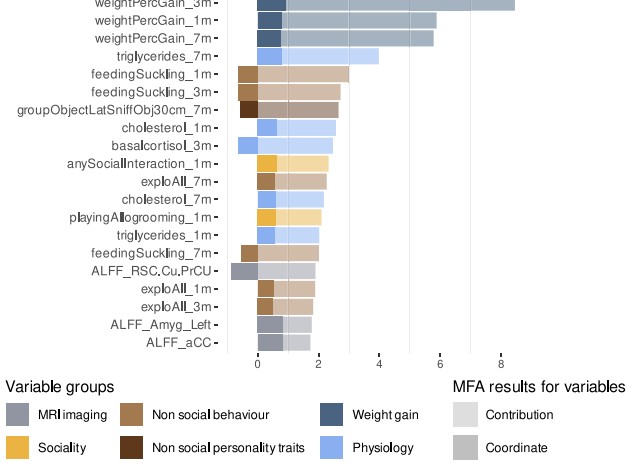

**Fig. 1 | Experimental design of our comprehensive and longitudinal study to assess how maternal presence beyond 6 months of age influences brain, behavioural, and physiological development. a** Timeline and **b** overview of the experimental protocol, with showing the main categories of the recorded parameters over time, including **c** brain development (functional/microstructural MRI brain metrics), **d** behavioural development (behavioural observations and tests), and **e** physiological development (hormonal and metabolic profiling). A comprehensive visualisation of the dataset is shown in (**f**) using an integrative multiple factor analysis, where ((**f**), left panel) shows the distribution of individuals along the first two axes of the MFA, and ((**f**), right panel) displays the variables that contribute the most to the first MFA axis, which discriminates between the '*with mother*' (orange) and '*without mother*' (purple) animals. Source data for panel (**f**) are provided as a Source Data file.

## Prolonged maternal presence shapes the brain microstructural organisation

Using our TEBTA resources (Fig. 2), we investigated the effects of prolonged maternal presence on the brain microstructural organisation in the offspring. We first evaluated the difference in grey matter concentration (GMC) using a voxel-based morphometry approach (VBM). VBM is a thoroughly validated technique for analysing images that provides an unbiased and comprehensive assessment of anatomical differences throughout the grey matter in both animal and human brains[23–26]. Using MRI $T_{1w}$ anatomical acquisitions, we estimated the GMC maps, which provided a global estimation of the GMC of the brain for each animal. Comparisons of GMC maps

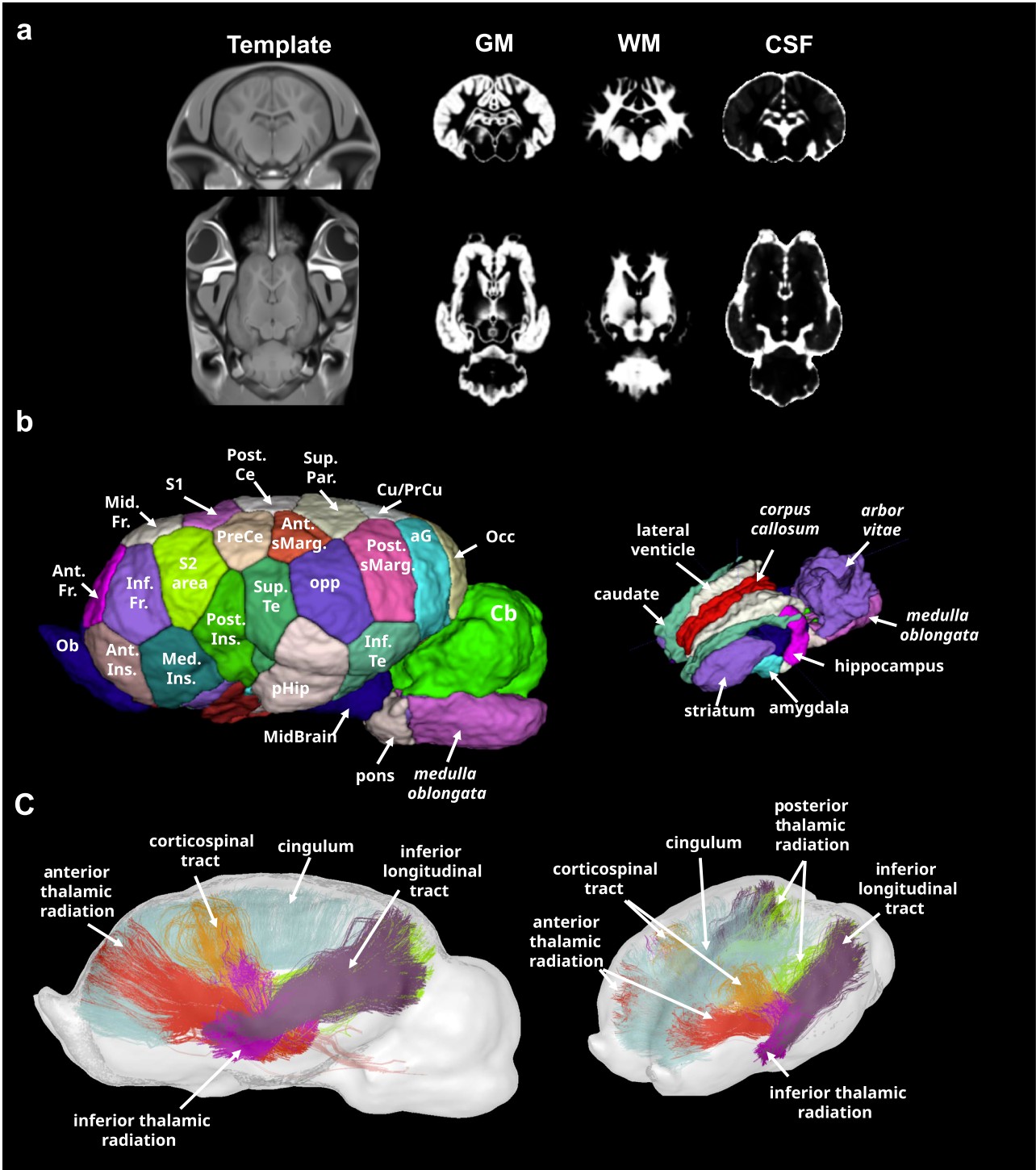

**Fig. 2 | The Turone equine brain template and atlas. a** Sagittal and axial brain slices of the Turone equine template, along with its associated tissue probability maps of grey matter (GM), white matter (WM) and cerebrospinal fluid (CSF). **b** Lateral and top views of the Turone equine brain atlas show the results of the multi-animal dictionary learning statistical approach for cortical parcellation using functional imaging data (left panel) and the results of the manual segmentation of the subcortical regions (right panel). **c** Lateral and top views of the Turone equine brain fibre atlas show the results of manual fibre clustering using diffusion imaging data. aG: angular gyrus, Ant. Fr.: anteromedial frontal region, Ant. sMarg: supramarginal region, Cb: cerebellum, Cu/PrCU: cuneus/precuneus, Inf. Te.: inferior temporal region, Inf. Fr.: inferior frontal region, med. Ins.: medial insular region, Mid. Fr.: middle temporal region, Ob: olfactory bulb, Occ: occipital region, Opp: parietal operculum, Phip: parahippocampal region, Post. Ce: postcentral region, Post. Ins: posterior insular region, Post. sMarg: posterior supramarginal region, Pre. Ce: precentral gyrus, S1: primary somatosensory area, S2: secondary somatosensory area, Sup. Te.: superior temporal region, Sup. Par.: supraparietal region.

revealed significantly lower GMC values (two-sample t test, voxel-level threshold $P < 0.05$, $t_{(21)} = 3.4414$, two-tailed, FDR-corrected) in the amygdala (amyg), the hippocampus (hippo), and the medial cingulate cortex (mCC) of and substantially higher values in both the thalamus (thal) and the hypothalamus (hypo) in animals belonging to the '*with mother*' group (Fig. 3). Although VBM is a highly sensitive method for studying changes in the brain microstructure, its specificity is limited, and the functional significance of such

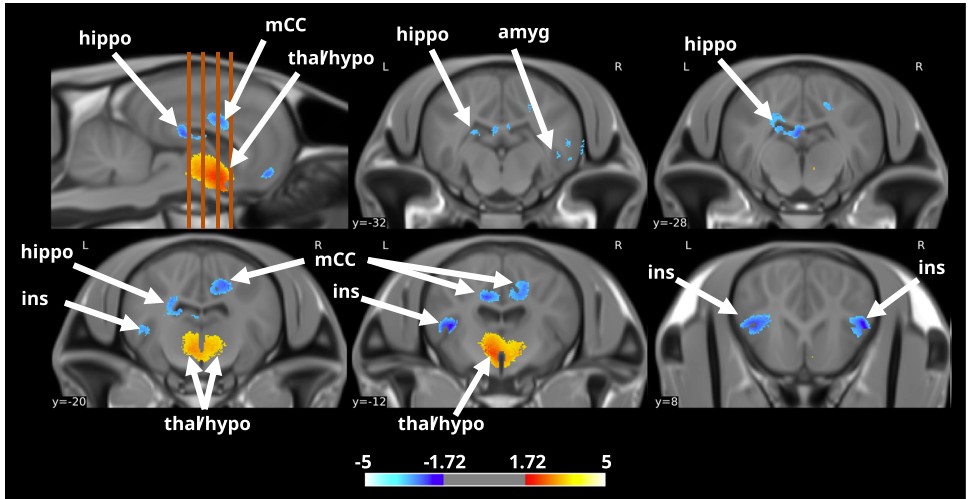

**Fig. 3 | Effect of prolonged maternal presence on the grey matter concentration in the equine brain (*with mother* > *without mother*).** Sagittal and axial brain slices showing differences in grey matter concentration (GMC) between groups. The orange dashed lines represent the approximate positions of axial brain slices.

The data were obtained from SPM unpaired two-sample t test analysis using a voxel-level threshold $p < 0.05$, t threshold (dof = 21) = 3.4414, two-tailed, FDR corrected, and a voxel threshold = 6.000. amyg: amygdala, hippo: hippocampus, hypo: hypothalamus, ins: insula, mCC: medial cingulate cortex, thal: thalamus.

microstructural modifications is still a matter of debate[27]. Therefore, we used a multishell 3D diffusion MRI approach (dMRI) optimised for neurite orientation dispersion and density imaging (NODDI) according to the model of Zhang et al. to better characterise these microstructural modifications[28]. From this imaging dataset, we extracted the fractional anisotropy (FA), along with three scalar parameters from the NODDI model: the intraneurite fraction (inf), also known as the neurite density index representing axons and possibly dendritic processes; the orientation dispersion index (ODI, where 0 indicates perfectly aligned straight axons and 1 indicates fully isotropic axons); and finally, the isotropic fraction (isoFrac, the extraneurite compartment, which encompasses all structures except neurites and free water, such as microglia, astrocytes, oligodendrocytes, neuronal somas, ependymal cells, extracellular matrices, and vascular structures)[29]. Comparisons of FA maps revealed significantly lower FA values (two-sample t test, voxel-level threshold $p < 0.05$; $t_{(21)} = 3.4414$; two-tailed, FDR corrected) within the anterior cingulate cortex (aCC), cuneus/precuneus (Cu/PrCu), pFC, and retrosplenial cortex (RSC) of animals in the '*with mother*' group (Fig. 4a). Notably, significantly higher ODI values (two-sample t test, voxel-level threshold $p < 0.05$, $t_{(21)} = 3.4414$, two-tailed, FDR-corrected) were observed in similar regions in animals belonging to the '*with mother*' group (Fig. 4b). Finally, significantly lower isoFrac values (two-sample t test, voxel-level threshold $p < 0.05$, $t_{(21)} = 3.4414$, two-tailed, FDR-corrected) were observed within the Cu/PrCu, the pFC and hypothalamus of animals belonging to the '*with mother*' group (Fig. 4c).

Taken together, both VBM and dMRI revealed that offspring benefiting from prolonged maternal presence exhibited significant microstructural modifications within pivotal brain structures, mainly those involved in physiological (thalamus) and behavioural (aCC/mCC, Cu/PrCu, pFC and RSC) regulation, or both (hypothalamus).

**Prolonged maternal presence shapes brain function**
We performed functional MRI (fMRI) optimised for a resting-state analysis to study the effects of the microstructural changes on brain function. From these data, we computed the fractional amplitude of low-frequency fluctuation (fALFF, 0.01–0.08 Hz) maps, a highly stable, specific and sensitive index of resting-state fMRI used to characterise spontaneous brain activity[30,31]. First, a comparison of fALFF revealed significant variations (two-sample t test, voxel-level

threshold $p < 0.05$, $t_{(21)} = 3.4414$, two-tailed, FDR-corrected) in spontaneous neuronal activity within the previously highlighted regions. Specifically, significantly lower fALFF values were observed within the Cu/PrCu, RSC and pFC of animals in the '*with mother*' group. Moreover, significantly higher fALFF values were observed within the aCC, PAG and amygdala (amyg) in the brains of animals in the '*with mother*' group (Fig. 5a–b).

This overlap between functional and microstructural modifications within the Cu/PrCu, the RSC, the pFC and the aCC, which are known to be play key roles in the DMN, a large-scale brain network involved in sociocognitive development in humans, suggests that prolonged maternal presence promotes the activity of this network in offspring[4,12]. We calculated the mean Pearson correlation coefficient between time series of spontaneous variations in BOLD signals extracted from the homologous regions involved in the DMN and DMN-like regions described in both humans and animals (aG, aCC/pFC and RSC) to evaluate the effect of prolonged maternal presence on functional connectivity within the DMN of foals. Using multiple regression analysis, we showed that spontaneous fluctuations in the BOLD signals in these regions were indeed highly correlated (voxel-level threshold $p < 0.001$, $t_{(21)} = 7.0544$, two-tailed, FDR corrected; Fig. 5c, left panel), revealing the presence of a strong, extensive brain network. Additionally, anticorrelated voxels were observed bilaterally within the somatosensorial cortical area (S1), which is consistent with previous descriptions of the DMN and DMN-like networks in humans, mice and rats. Consequently, we propose that the aG, aCC/pFC and RSC regions together form a DMN-like network in the domestic horse, akin to what has been described in other mammals. Finally, we performed a multiple regression analysis focused specifically on animals raised either with or without their mother to evaluate the function of the DMN in both experimental groups ('*with mother*' group; Fig. 5c, centre panel). An analysis of animals raised with their mothers revealed that the cluster size and spatial organisation of the DMN were similar to those observed in our population analysis. These results contrast with the results obtained with animals raised without mothers ('*without mother*' group, Fig. 5c, right panel), for which we observed a smaller DMN and a decrease in functional connectivity within the DMN in this population. These data suggest that the presence of the mother could be involved in the maturation of the DMN-like brain network of the offspring of domestic horses.

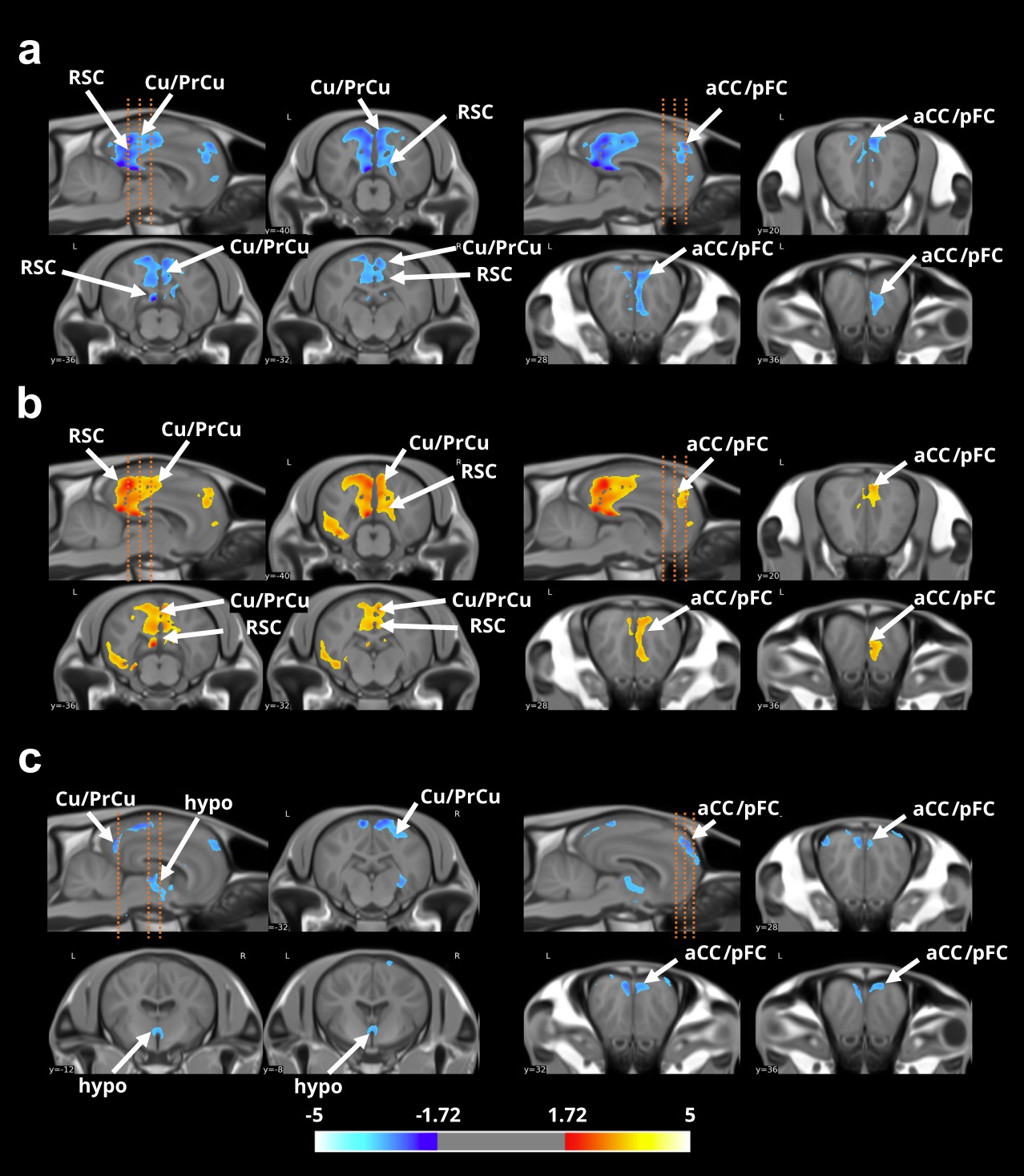

**Fig. 4 | Effect of prolonged maternal presence on the cytoarchitecture of the equine brain (*with mother > without mother*).** Sagittal and axial brain slices showing differences in (**a**) fractional anisotropy (FA), (**b**) the orientation dispersion index (ODI) and (**c**) the isotropic fraction between groups. The orange dashed lines represent the approximate positions of axial brain slices. The data were obtained from SPM unpaired two-sample t test analysis using a voxel-level threshold $p < 0.05$, t threshold (dof = 21) = 3.4414, two-tailed, FDR corrected, and a voxel threshold = 15.000. aCC: anterior cingulate cortex, Cu/PrCu: cuneus/precuneus, hypo: hypothalamus, pFC: prefrontal cortex, RSC: retrosplenial cortex.

## Prolonged maternal presence shapes behavioural development

Analyses of the percentage of scans spent in each behavioural activity combining all periods of observations (from the 1st month to the 7th month of prolonged maternal presence) using *Aligned Ranks Transformation ANOVAs* showed that animals belonging to the 'with mother' group displayed higher levels of exploration ($F_{(1,22)} = 11.74$, $p = 0.002$), rest ($F_{(1,22)} = 6.64$, $p = 0.02$), social interaction (for any type: $F_{(1,22)} = 19.58$, $p < 0.001$; and for affiliative interactions: $F_{(1,22)} = 16.10$, $p < 0.001$) and less time spent feeding ($F_{(1,22)} = 29.79$, $p < 0.001$) (Fig. 6a). Suckling was included in the total feeding time to ensure a relevant comparison between experimental groups. For foals in the 'with mother' group, the percentage of scans showing suckling was

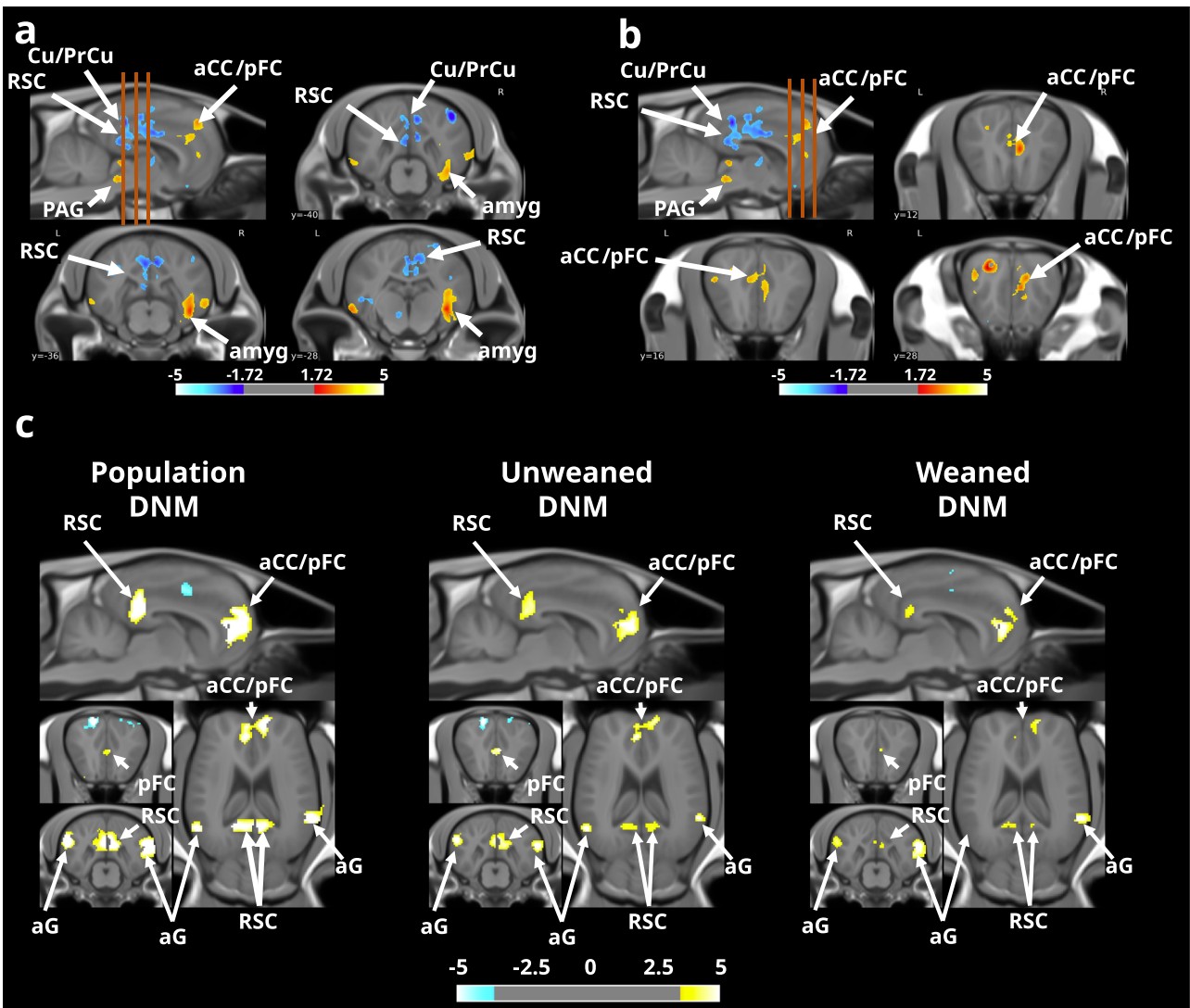

**Fig. 5 | Effect of prolonged maternal presence on the functional organisation of the equine brain (*with mother* > *without mother*).** Sagittal and axial brain slices showing changes in the amplitude of low-frequency fluctuations (ALFFs) in the posterior (**a**) and anterior (**b**) regions between groups. The orange dashed lines represent the approximate positions of axial brain slices. The default mode network (DMN) is identified in the domestic horse by calculating the mean Pearson correlation coefficient between time series of spontaneous variations in blood oxygen level-dependent (BOLD) signals extracted from anatomically defined regions of interest (ROIs), which are functionally homologous to regions involved in the DMN in numerous species (aG, aCC/pFC and RSC) and in which spontaneous variations in the BOLD signal are highly correlated at rest between them. In (**a**, **b**), the orange dashed lines represent the approximate positions of axial brain slices. The data were obtained from SPM unpaired two-sample t test analysis using a voxel-level threshold $p < 0.05$, t threshold (dof = 21) = 3.4414, two-tailed, FDR corrected, and a voxel threshold = 15.000. For **c**, the data were obtained from SPM multiple regression analysis using a voxel-level threshold $p < 0.001$, t threshold (dof = 21) = 7.0544, two-tailed, FDR corrected, and a voxel threshold = 500. aG: angular gyrus, aCC: anterior cingulate cortex, amyg: amygdala, Cu/PrCu: cuneus/precuneus, PAG: periaqueductal grey, pFC: prefrontal cortex, RSC: retrosplenial cortex.

$2.08 \pm 0.36\%$ at 1 month after weaning, $2.19 \pm 0.55\%$ at 3 months after weaning, and $2.83 \pm 0.70\%$ at 7 months after weaning (means ± SEs). No foals belonging to the '*without mother*' group were observed suckling. Despite accounting for suckling in the calculation of total feeding time, foals in the '*with mother*' group still spent significantly less time feeding overall than foals in the '*without mother*' group.

A social network analysis based on spatial proximity matrices (i.e. less than 1 m between two animals) was used to calculate individual centrality scores[32]. Animals belonging to the '*with mother*' group had higher centrality scores ($F_{(1,22)} = 12.50$, $p = 0.002$). Interestingly, foals did not differ in the percentage of time spent isolated (i.e. more than 10 m from any other group member, $F_{(1,22)} = 0.21$, $p = 0.65$). Behavioural tests were conducted at +7 months after maternal separation to experimentally assess foals' reactivity profiles in the social dimension (sociability test), non-social dimension (novel object test individually and novel surface test) and both dimensions together (novel object test in group). The results showed that foals raised with their mother approached an unfamiliar horse more rapidly during the sociability test and required fewer training sessions to reach compliance for care training (Fig. 6b).

**Prolonged maternal presence shapes physiological development**
The temporal evolution of the effect of the prolonged presence of the mother on the physiological parameters of the offspring was tested using aligned rank transformation ANOVA and post hoc tests (Fig. 7). Overall, the weight gain ($F_{(1,22)} = 56.66$, $p < 0.001$) was greater for animals in the '*with mother*' group, although they spent less time feeding; these animals presented higher concentrations of triglycerides ($F_{(1,22)} = 5.82$, $p = 0.02$) and cholesterol ($F_{(1,22)} = 11.74$, $p = 0.002$) and lower cortisol concentrations ($F_{(1,22)} = 11.74$, $p = 0.002$). Greater weight gain was observed at every time point (at +1 and +3 months:

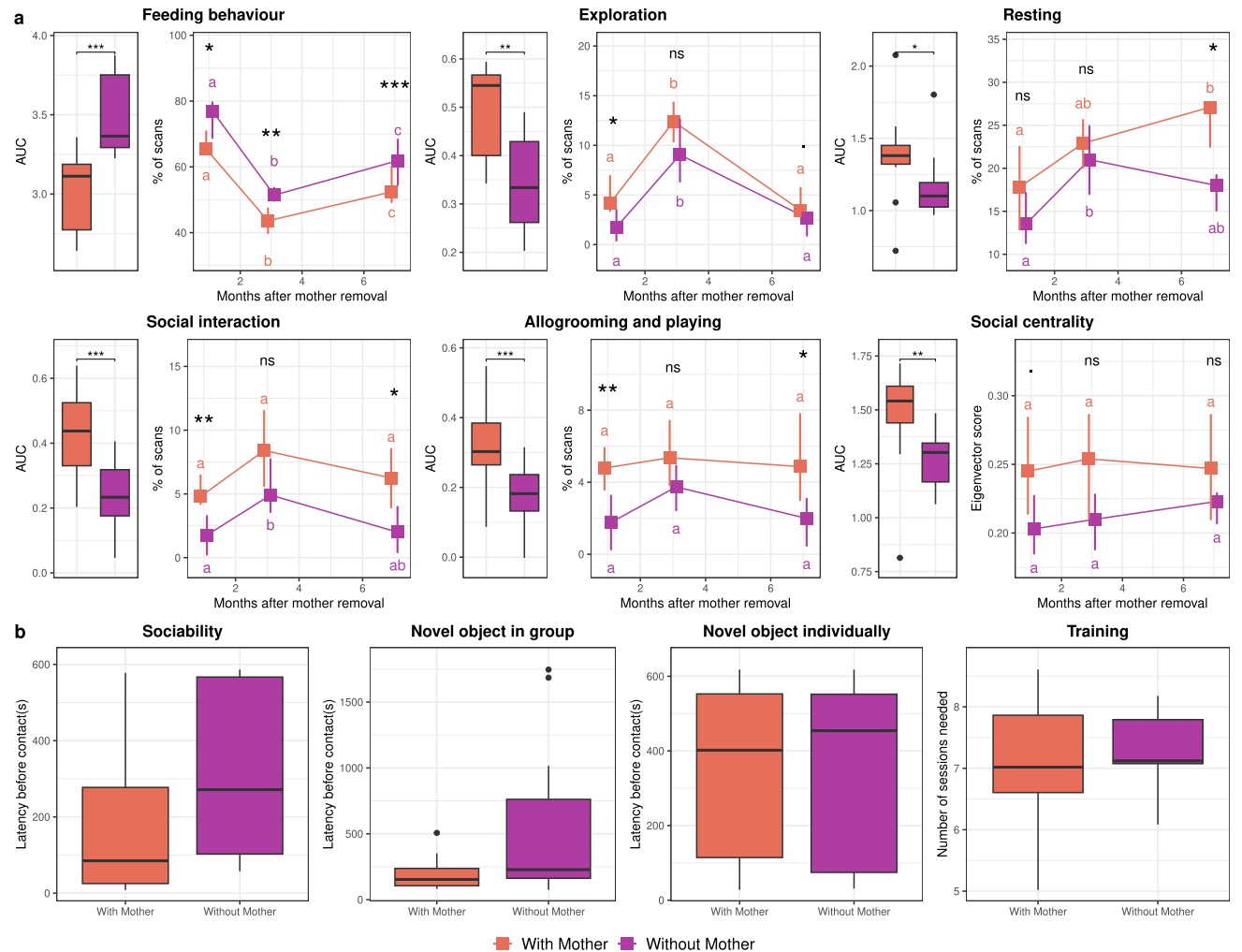

**Fig. 6 | Prolonged maternal presence promotes sociability, exploration, and feeding efficiency.** Longitudinal observations from +1 to +7 months after weaning. Boxes represent the medians, with error bars indicating the 95% confidence intervals of the median (n = 12 foals separated by groups; orange indicates foals with their mother and purple indicates foals without their mother) (**a**). Medians sharing the same letter are not significantly different from each other. Stars indicate significant differences between foals with or without their mothers (*** p < 0.001, ** p < 0.01, * p < 0.05, ns nonsignificant). *Aligned ranks transformation ANOVAs* were used to test differences at different time points and between foals with or without their mothers. Behavioural tests conducted at +7 months after weaning (**b**). The variables presented were selected based on a multivariate analysis. Boxplot whiskers extend to ±1.5× the interquartile ranges for each treatment. Box plots show the distribution of data: the centreline represents the median, the box bounds the 25th and 75th percentiles (interquartile range, IQR), and the whiskers extend to the most extreme data point within 1.5 × IQR from the box. Data points beyond the whiskers are considered outliers. Source data, including all p values, are provided as a Source Data file.

t(48) = 6.01 and t(48) = 6.72, p < 0.001; at +7 months: t(48) = 4.08, p = 0.002). The higher level of triglycerides in the 'with mother' group was more pronounced at +1 month (t(62) = 2.93, p = 0.04) and +7 months after the beginning of treatment (t(48) = 3.62, p = 0.005), whereas the higher level of cholesterol was more pronounced at +1 month (t(58) = 4.25, p < 0.001) and +7 months (t(58) = 3.31, p = 0.01). The lower levels of cortisol appeared to be restricted to +3 months after maternal separation (t(65) = −3.13, p = 0.02). No differences were observed in oxytocin, glycaemia, IGF-1 or insulin levels.

### Relationships between brain structure and function and biobehavioural trajectories

The correlation structure among the six data groups (*weight gain, non-social behaviour, non-social personality traits, sociality, MRI,* and *physiology*) was investigated using multiblock projection to latent structures (PLS) models with sparse discriminant analysis (multiblock PLS-sDA). This comprehensive multivariate approach optimises correlations across various data groups and conducts discriminant analysis to identify a specific signature for each experimental group. The selected final model is presented in Fig. 8. The hues and ellipses associated with the treatment demonstrate the ability of each component to distinguish between the presence and absence of the mother (Fig. 8a). While the centroids of each treatment are clearly separated, a moderate degree of overlap exists between the sample groups within their confidence ellipses. Furthermore, the first components from each data group were correlated with at least one other group (indicated by the large correlation coefficients in the bottom left, Fig. 8a). Our final statistical model revealed that the first components of the 'MRI' variable were significantly correlated with 'sociality' (r = 0.79), 'non-social behaviours' (r = 0.87), 'weight gain' (r = 0.87) and 'physiology' (r = 0.9). Notably, a redundant pattern of antagonistic correlations between the functional activity of aCC and RSC-Cu/PrCu (ALFF) was observed for social interaction and social isolation at +1 month, playing and allogrooming (i.e. affiliative interactions) at +1/+7 months, feeding behaviours at +1/+3/+7 months, weight gain at +1/+3 months, cholesterol levels at +1/+7 months, and triglyceride levels at +7 months.

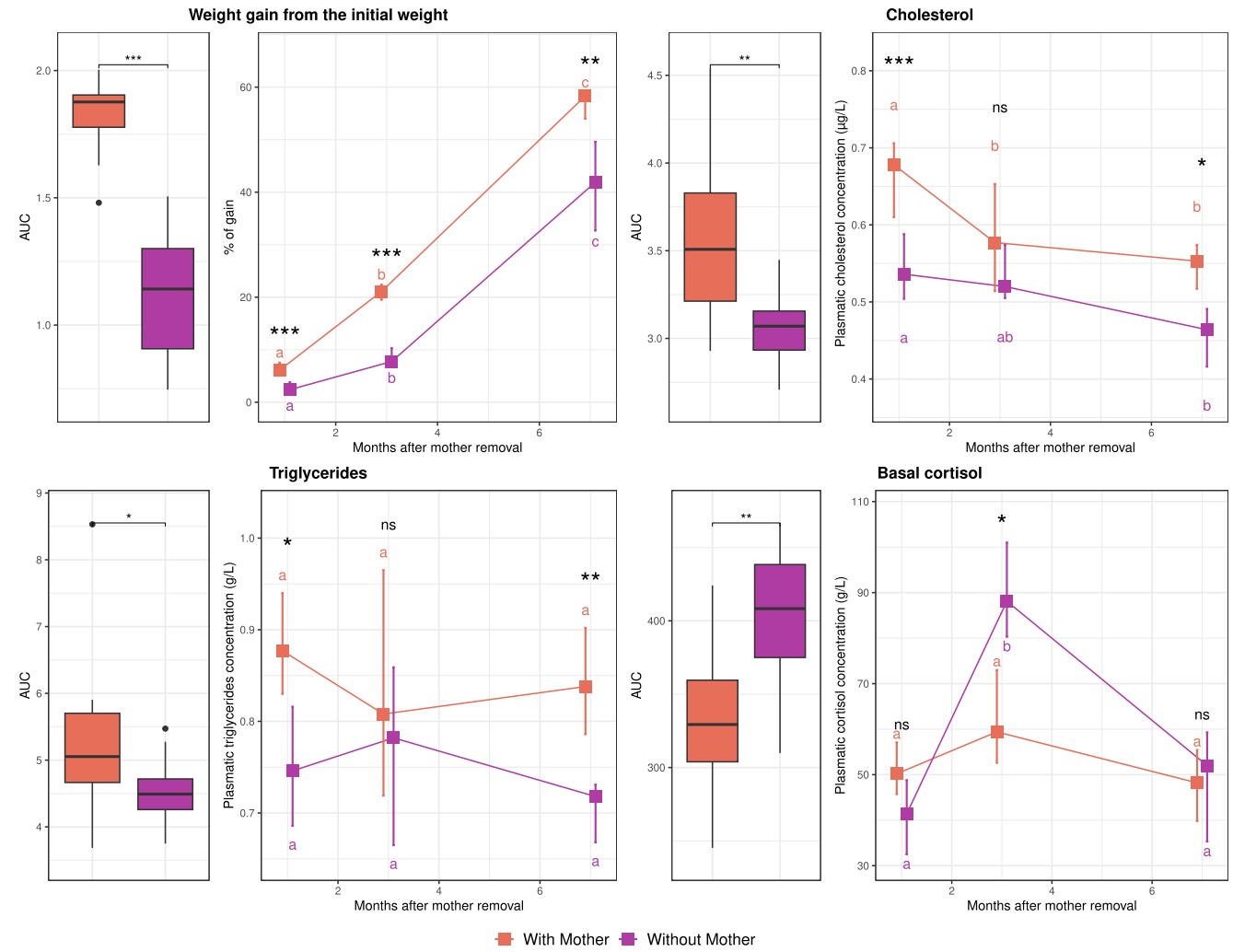

**Fig. 7 | Prolonged maternal presence influences weight gain and changes in basal plasma cortisol, cholesterol and triglyceride concentrations over time.** The boxes indicate the medians. The error bars indicate the 95% confidence intervals of the medians ($n = 12$ foals per group; orange indicates foals with their mother and purple indicates foals without their mother). Medians with the same letter are not significantly different. Stars represent significant differences between foals with or without their mothers (*** $p < 0.001$, ** $p < 0.01$, * $p < 0.05$, $p < 0.1$, ns nonsignificant). *Aligned ranks transformation ANOVAs* were used to test differences at different time points and between foals with or without their mothers. Box plots show the distribution of data: the centreline represents the median, the box bounds the 25th and 75th percentiles (interquartile range, IQR), and the whiskers extend to the most extreme data point within $1.5 \times IQR$ from the box. Data points beyond the whiskers are considered outliers. Source data, including all $p$ values, are provided as a Source Data file.

Additionally, the functional activity of the amygdala (left and right ALFF) was strongly correlated with the same variables and showed a similar direction to the functional activity of the aCC. Finally, microstructural modifications (VBM) within the mid-cingulate cortex (mCC), hippocampus and insula (left and right) were negatively correlated with +1/+3-month weight gain and +7-month triglyceridaemia (Fig. 8b). In conclusion, our model shows that animals belonging to the '*with mother*' group display higher functional activity within both the aCC and the amygdala (left/right) and are more social and less isolated up to 7 months after treatment. In contrast, animals belonging to the '*without mother*' group display higher functional activity of the RSC-Cu/PrCu and higher GMC values within the hippocampus and mCC and spend more time feeding without gaining more weight (Fig. 8c).

## Discussion

Mapping the associations among the brain, behaviour, and physiology is a critical objective in neuroscience. In this context, multivariate methods have gained popularity, and latent structures (PLS) models have emerged as tools for identifying complex brain-behaviour-physiology associations, providing a better understanding of how structural and functional brain organisations are linked with physiology and/or behaviours[33]. These advanced statistical approaches enabled us to combine brain imaging metrics with longitudinal biobehavioural assessments in foals. They revealed that social affiliative behaviours during a mid-childhood-like stage foster the development of the allostatic regulatory system in offspring through two fundamental biological dimensions: social life and feeding/metabolic efficiency.

First, our results support that affiliative behaviours play a critical role in shaping social trajectories. During the mid-childhood-like stage, the functional brain metrics of RSC-Cu/PrCu and aCC/pFC, two major hubs of the DMN, in offspring were shown to be influenced by social interactions with caregivers in our statistical models. Specifically, offspring benefiting from prolonged maternal presence during this critical period exhibited lower RSC-Cu/PrCu activity but higher aCC/pFC activity. In comparison, human brain imaging studies have reported distinct roles for these two DMN hubs in social behaviour. The RSC-Cu/PrCu region (the PCC in humans) is involved primarily in self–other distinctions, whereas the aCC/pFC region is associated with monitoring both one's own mental states and others' mental states. Moreover, the DMN plays a critical role in self-reference,

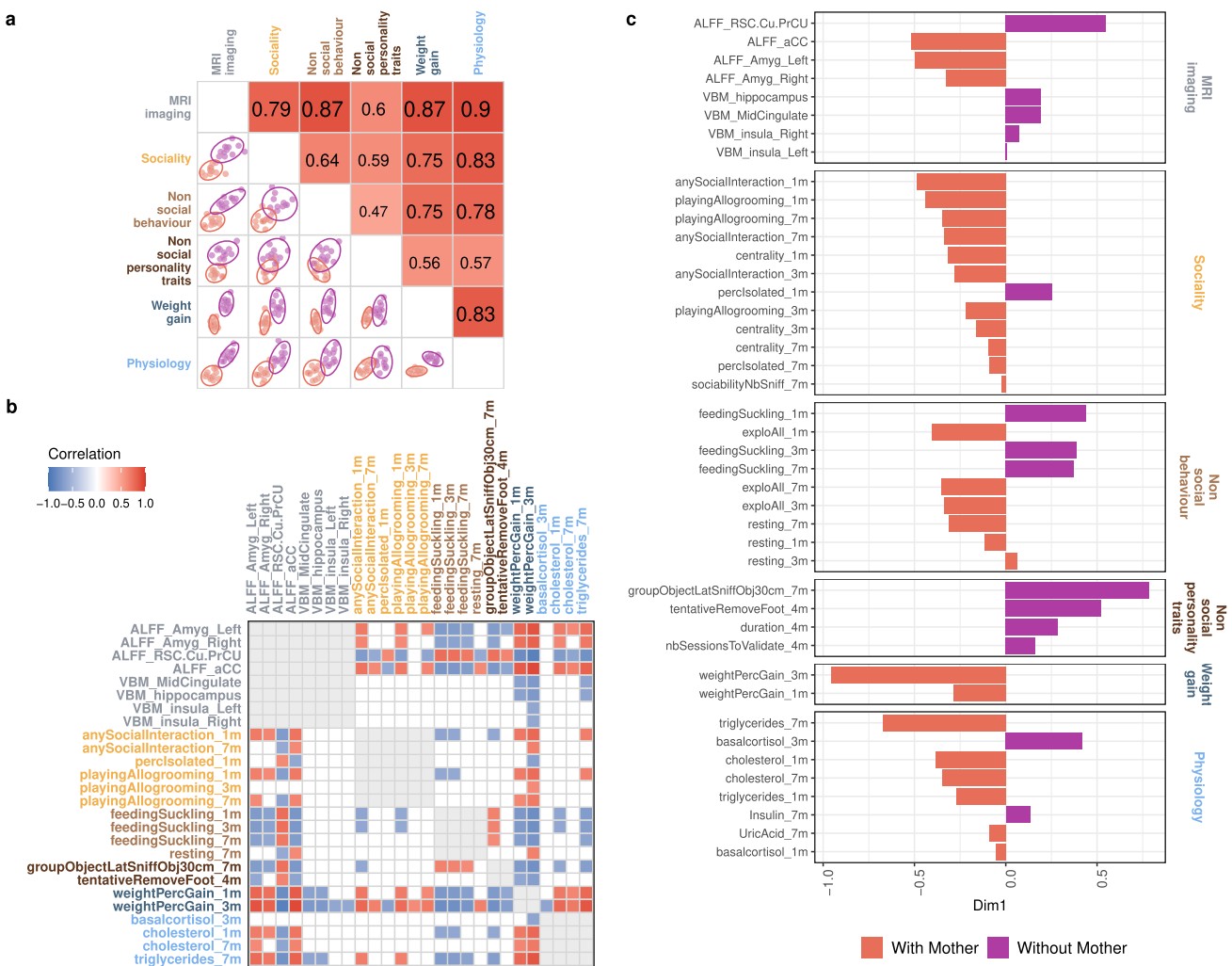

**Fig. 8 | Multiblock sPLS-DA identifies differences between foals and correlations between variables.** Correlation between the first dimensions of each group of variables (**a**). Correlations between variables belonging to two different groups (**b**). Loading weights of each selected variable on each dimension and each data group (**c**). The colour indicates the class in which the variable has the maximum level of expression using the median (orange represents foals with their mother and purple represents foals without their mother). The most important variables (according to the absolute value of their coefficients) are ordered from top to bottom. The following abbreviations were used: ALFF: amplitude of low-frequency fluctuations, VBM: voxel-based morphometry, ODI: orientation dispersion index, aCC: anterior cingulate cortex, RSC: retrosplenial cortex, Cu/PrCu: cuneus/precuneus, Amyg: amygdala, 1 m/3 m/4 m/7 m: measurements performed at 1, 3, 4 or 7 months after maternal separation, anysocialinteraction: percentage of scans in which foals spent time in any social interaction, playingallogrooming: percentage of scans in which foals spent playing or allogrooming, centrality: social network

centrality score, percIsolated: percentage of scans in which foals spent time isolated from conspecifics, sociabilityNbSniff: number of social sniffs during the sociability test, feedingSuckling: percentage of scans in which foals spent time feeding (including suckling), expoAll: percentage of scans in which foals spent time exploring the environment, resting: percentage of scans in which foals spent time resting, groupObject.atSniffObj30cm: latency to sniff a novel object from a distance shorter than 30 cm during the group novel object test, tentativeRemoveFoot: tentative hoof lifting during the farriery session, nbSessionsToValidate: number of training sessions required to reach the compliance criterion during training, weightPercGain: percentage of weight gain, triglycerides: plasma triglyceride concentration, cholesterol: plasma cholesterol concentration, basal cortisol: basal plasma cortisol concentration, insulin: plasma insulin concentration, uric acid: plasma uric acid concentration, oxytocin: plasma oxytocin concentration. Source data are provided as a Source Data file.

episodic and autobiographical memory, theory of mind, and other high-level social cognitive functions in humans[34]. While the DMN has been previously identified in a few animal species (e.g. mice, rats, marmosets and chimpanzees)[35], our study provides evidence for a DMN-like network in horses. Furthermore, our results highlight that this network, which is central to complex social behaviours in humans, may also support optimal social development in domestic horses. Indeed, our results also revealed a statistically significant association between the functional activity of these DMN regions and foals' social development. Offspring benefiting from prolonged maternal presence exhibited more positive and adaptive social behaviours, alongside increased exploratory and resting behaviours.

Notably, these developmental advantages emerged even though all young individuals were raised in the same socially rich environment, highlighting the specific role of the mother–offspring bond in shaping these trajectories. In addition to the DMN, we also observed functional modifications within the amygdala associated with social behaviours in foals. The amygdala is a hub structure involved not only in social processing but also in a large variety of behavioural processes, including fear- and pain-related processes[36,37]. Through its reciprocal connection with the aCC, a highly conserved brain network across mammalian evolution[38], the amygdala has been shown to influence social decision-making behaviours in monkeys[39]. Indeed, when monkeys choose to deliver juice rewards to a conspecific

(positive other-regarding preference), increased coherence is observed between the spiking of neurons in the amygdala and beta oscillations of the aCC but also between the spiking of aCC neurons and gamma oscillations in the amygdala, whereas when monkeys choose to deliver a juice reward for themselves rather than a conspecific (negative other-regarding preference), this coherence vanishes[39]. This interareal oscillatory coordination between the aCC and amygdala suggests that this circuit controls social behaviours. Thus, the circuitry is highly sensitive to early-life stressors[40], and impaired functioning of this pathway could promote a deficit in social skills[39]. Here, our model shows that animals belonging to the 'with mother' group display higher functional activity within both the aCC and the amygdala (left/right) and are more social and less isolated, demonstrating that affiliative behaviours should impact the development of functional coordination between the aCC and the amygdala. Taken together, our findings underscore the essential role of prolonged maternal presence in fostering the successful development of social competences and suggest that affiliative behaviours are involved in the establishment of foundational networks involved in social development across social mammalian species[41].

Second, our study suggests that caregiver-driven maturation supports the physiological control of feeding and metabolic trajectories. Indeed, consistently throughout our study, compared with those separated from their mothers, foals benefiting from prolonged maternal presence spent less time feeding and displayed higher levels of exploration and rest. Interestingly, they also experienced greater weight gain and presented higher concentrations of triglycerides and cholesterol, consistent with previous findings in this species[7]. Because these physiological parameters are intimately linked to feeding efficiency, metabolism, and energy storage, these results highlight a certain advantage in the metabolic and growth statuses of foals that benefit from prolonged maternal presence. First, lipid metabolism is essential for the development of the central nervous system, as cholesterol is a primary component of neuronal membranes and intracellular structures. Numerous longitudinal and cross-sectional studies have linked low blood cholesterol and triglyceride levels to impaired cognition, recognition memory, and inhibitory processing in humans[42]. In the present work, offspring with prolonged maternal presence preserved higher cholesterol and triglyceride levels and achieved better weight gain while they spent less time feeding, suggesting that they maintained more efficient feeding behaviours and/or metabolic functioning. Several mechanisms could account for this metabolic efficiency. Access to maternal milk, although it represented only ~2.5% of the foals' total activity budget, may still have provided valuable nutrients that are beneficial for growth. The benefits of lactation are well established in humans, as prolonged breastfeeding is advantageous for both physiological and cognitive development[43]. Although both experimental groups consumed primarily solid diets, even minimal maternal lactation might have conferred some physiological benefits for these animals. Additionally, feeding is a goal-directed behaviour essential for the survival of all metazoan organisms and is believed to be rooted within innate neuronal circuits, particularly within the hypothalamus[44]. In our study, we detected differences in the microstructural organisation of the hypothalamus between the two groups, confirming that fundamental feeding pathways might be influenced by prolonged maternal presence. Moreover, social learning and emotional regulation could further refine feeding behaviour through additional, higher-order regulatory mechanisms. Indeed, we found that functional brain metrics in the aCC/pFC and RSC-Cu/PrCu (DMN) were associated with feeding behaviour, suggesting a role for higher-order cognitive processes in regulating food intake. Consistent with these findings, research in humans has revealed alterations in the DMN of patients with eating disorders (e.g. *anorexia nervosa*), linking disruptions in cognitive control networks to feeding

disorders[45]. Our statistical model also indicated a significant association between functional activity of the amygdala and feeding behaviour, consistent with the findings of previous studies in mice showing that specific neuronal populations in the central nucleus of the amygdala integrate negative affective state signals, such as fear/anxiety, and impact feeding behaviours[46]. Prolonged maternal presence could limit these negative affective states through social buffering mechanisms, as suggested by the lower cortisol levels observed after 3 months of prolonged maternal presence, ultimately supporting the development of more efficient feeding behaviours in offspring.

Ultimately, the present study allowed for the implementation of dedicated resources for in vivo MRI investigations of the horse brain. Indeed, the availability of MRI-compatible population brain templates and atlases is mandatory for studying the brains of large animal models in neuroscience, for which the scientific community increasingly advocates[47–49]. Together, templates and atlases enable group analyses to address the size and volume of pivotal brain structures and reveal both functional and structural brain networks[50], as well as the brain microstructural organisation in vivo[28]. Among the animal models used in neuroscience, the domestic horse is of special interest because of its remarkable cognitive performance[51,52]. Indeed, horses exhibit testable cognitive functions and can perform learning, discrimination, match-to-sample and memory tasks[53]. Additionally, the horse is the longest-lived of the common domestic species (25–30 years), making this animal model particularly relevant for the study of progressive neurological diseases[51,54]. Hence, in the first step of our study, we generated a set of neuroinformatic tools that provide a comprehensive resource dedicated to MRI studies of the horse brain in vivo, namely, a high-resolution brain template and its associated GM, WM and CSF priors built from 23 individuals for anatomical, functional, and diffusion imaging analyses in horses. We also provided an in vivo horse brain atlas composed of a mosaic of 115 regions of interest (ROI), which offers accurate segmentation of both cortical and subcortical areas. Finally, our study provides a large range of acquisition protocols for in vivo MRI scanning in foals to the community (see Supplementary Figs. 3–9 and Supplementary Table 2 for details and quality checks). The protocols provided here could be considered useful starting points for research/veterinary centres interested in imaging the brains of Equidae. Nevertheless, our sequences have significant room for improvement (TR of functional data, resolution of diffusion data, contrast of anatomical data), and a multisite-based strategy will be necessary to test our protocols and improve their quality. This comprehensive set of MRI-compatible tools will allow standardisation of horse brain MRI data analysis and may facilitate the development of multicentric preclinical studies in horses, as previously described for rats[55].

In conclusion, despite being housed in a socially enriched environment with age-matched peers and adult conspecifics, the neurological, behavioural and physiological development of foals are impaired when they are coping with the absence of their mothers. This result suggests the presence of a developmental control system nested at the social level in horses, a theory previously proposed for rodents, monkeys, and humans[1,2,4,20]. In humans, Atzil et al.[4] proposed a theoretical framework where parents provide social cues (care) that foster the development of the allostatic regulatory system via the maturation of whole-brain neural networks, allowing for balanced biobehavioural development until adulthood. Essentially, the allostatic regulatory system is responsible for the maintenance of physiological and behavioural states through predictive and anticipatory regulation. This system must be fuelled by rich social interactions for the building of conceptual models from the detection of statistical regularities in interoceptive and exteroceptive inputs to generate predictions. The acquisition of these conceptual models is associated with the maturation of domain-general core networks in which interoceptive

and exteroceptive inputs are constantly integrated to construct models. This theoretical framework is consistent with our observations, as the absence of the mother reduced the size and functionality of the DMN in offspring in association with decreased sociality and impaired feeding behaviours and lipid metabolism. Our observations suggest that the theoretical framework proposed by Atzil et al.[4] for humans could be extended to horses and perhaps to numerous social mammals. Nevertheless, an alternative hypothesis could posit that the presence of the mother might have provided indirect benefits through pre-existing affiliative bonds within the group. These additional social resources may have further amplified the contrast with weaned individuals, increasing the level of social challenge for weaned horses compared with foals benefiting from the presence of their mothers. Future studies using focal sampling or continuous recording to capture overall social interactions will be necessary to elucidate the affiliative dynamics and disentangle the respective contributions of maternal presence and group-level affiliative structure to better understand how prolonged maternal presence shapes developmental trajectories. Future investigations will also need to determine whether the changes detected in brain function caused by maternal loss are responsible for the physiological changes observed in foals or if the physiological modification leads to altered brain function. To date, both hypotheses remain possible. Multiple imaging sessions, long-term behavioural testing, and closer physiological monitoring will be necessary to clarify which changes (brain, behaviour or physiology) occur first to determine the directionality of the correlations detected here. By coupling these discoveries with dedicated MRI-based resources, our study positions the horse as a relevant large-mammal model for investigating extended caregiver–offspring relationships. These insights also invite us to reconsider weaning practices for animals under human care, as preserving maternal contact may yield significant long-term benefits for both animal welfare and developmental trajectories.

## Methods

### Subjects

The subjects were 24 Welsh foals, 12 females and 12 males, aged $6.64 \pm 0.07$ months (mean ± SE), and their 24 mothers were aged $8.2 \pm 0.3$ years on the day of the removal of the mothers. All the animals (foals and mothers) were born and housed at the Animal Physiology Experimental Unit (UEPAO, INRAE, 37380 Nouzilly, France). They all lived in groups in, depending on the season, indoor collective stalls ($20\,m \times 25\,m$) on straw with free access to an outdoor paddock (from November 26, 2021, to April 23, 2022) or in large outdoor grass pastures (April 23, 2022, to June 27, 2022). The dietary ration, which was primarily based on *ad libitum* access to forages, pasture and mineral block, was designed to meet the theoretical nutritional requirements (INRA 2011 tables) for weaned foals of this type and age[56]. Foals were born between May 13 and June 27, 2021. During the first five months of life, they were all kept with their mothers in the same outdoor pasture. In October 2020, two distinct herds were formed and kept in two different pastures: 12 dyads of male foals/mothers (Herd A) and 12 dyads of female foals/mothers (Herd B). The protocol started in November 2020 (Fig. 1a). Both sexes were included in the study, with balanced numbers of males and females in each group. However, the effect of sex could not be statistically tested independently, as sex was confounded by the housing conditions: males and females had to be housed separately for ethical reasons to avoid the risk of reproduction among young animals. Sex and group information are nevertheless indicated in the figures (e.g. Fig. 1f) for transparency.

All procedures were performed in accordance with the European directive 2010/63/EU for animal protection and welfare used for scientific purposes and approved by the local ethical committee for animal experimentation (CEEA VdL, Tours, France, ref. APAFIS #32985-2021091516064302).

### 'With mother' and 'Without mother' treatments

Half of the subjects were assigned to the 'with mother' treatment ($N = 12$: 6 males and 6 females), and the other half were assigned to the 'without mother' treatment ($N = 12$: 6 males and 6 females). The 'with mother' and 'without mother' foals were balanced according to the date of birth, weight and paternal origin. The 'without mother' treatment consisted of removing half of the mothers from each group (A and B) simultaneously when the foals were $6.64 \pm 0.07$ months old (mean ± SE). Consequently, after maternal separation, each herd (A and B) was composed of 6 without-mother foals, 6 with-mother foals, and their 6 adult mothers. Therefore, the presence or absence of the mother was the sole difference between the groups. Foals were kept within the same social herd (A or B) and could therefore interact with each other and with the adult mothers that were still present. One female foal belonging to the 'with mother' group left the experiment 4 months after the maternal separation procedure for medical reasons. Imaging, behavioural and physiological data from this animal were retrieved for the final analysis.

### Magnetic resonance imaging

**Anaesthesia.** Imaging procedures were implemented in the PIXANIM imaging facilities, which are located on the same site as the animal housing facility. The complete procedure is described in the Supplementary Information and was inspired by our previous study[57]. Briefly, anaesthesia was induced in two successive stages. First, a dose of romifidine hydrochloride was injected intravenously as a premedication. Five minutes later, ketamine hydrochloride was injected intravenously to induce anaesthesia. The animal was placed in the supine position on the MRI table, a saphenous catheter and a urinary drain were then placed, and an ophthalmic ointment was applied to avoid ocular dryness. A veterinarian and an animal technician were constantly present to monitor the animals' respiratory and cardiac pulses. The mean duration of anaesthesia from induction to the last injection was approximately 3 h.

**In vivo MRI acquisitions.** MRI data were acquired on 3 Tesla VERIO Siemens systems (Erlangen, Germany) using two flexible coils (Siemens FLEX Large 4 elements) tied around the head, and three MR image sequences were acquired for each animal. MR sequences were optimised to be performed in a time compatible with anaesthesia (approximately 3 h), reduce artefacts (folding, truncation, etc.), and optimise the SNR. With respect to these specifications, the following sequences were used:

For the brain morphometry investigations, a three-dimensional $T_{1w}$ MPRAGE acquired in the coronal plane was used with the following parameters: echo time/repetition time = 2.67 ms/2500 ms, flip angle = 12°, inversion time = 900 ms, number of excitations = 3, partial Fourier transform = 1, slice thickness = 1 mm, slice number = 208, field of view = $256 \times 256$ mm, matrix = $256 \times 256$, and final resolution = 1 mm³.

We investigated the brain microstructure using a diffusion-weighted MRI protocol based on a two-dimensional $T_{2w}$ spin–echo sequence acquired in the axial plane over 3 different shells optimised for the NODDI model using the following parameters: Shell 1−b = 300 s/mm², 6 directions; Shell 2−b = 700 s/mm², 30 directions; and Shell 3−b = 2000 s/mm², 64 directions. The fixed parameters were echo time/repetition time=109 ms/11.5 s, flip angle = 90°, number of excitations = 1, partial Fourier transform = 0.75, slice thickness = 2.4 mm, slice number = 57, field of view = $256 \times 256$ mm, matrix = $128 \times 128$, final resolution $2 \times 2 \times 2.4$ mm³, and one $b = 0$ per shell. Sequences were acquired in different reading phases (left-right and right-left) for distortion correction.

Brain function was investigated using a $T_{2w}$ spin–echo–planar imaging (SE–EPI) sequence acquired in the axial plane and left/right

reading phase with the following parameters: echo time/repetition time = 24 ms/3.97 s, flip angle = 90°, number of excitations = 1, partial Fourier transform = 1, slice thickness = 3.3 mm, slice number = 40, field of view = 220 × 220 mm, matrix = 110 × 110, final resolution = 2 × 2 × 3.3 mm$^3$, and number of repetitions = 250. Additionally, two similar sequences in different reading phases (left-right and right-left) of ten volumes each were acquired for distortion correction. Only one case of myositis was detected after the imaging procedure; nevertheless, the animal fully recovered within one week. DICOM data were converted to NIFTI format and organised as standardised datasets according to the Brain Imaging Data Structure (BIDS) using BIDScoin and are downloadable on Zenodo.

**TEBTA template creation.** T$_{1w}$ MPRAGE data acquired for each animal were corrected for noise and signal bias using Ginkgo and N4BiasFieldCorrection, respectively, and coregistered to the template reported by Johnson et al.[51] using antsRegistrationSyNQuick. Coregistered data were used to segment each brain using SPM and the GM, WM, and CSF priors provided by Johnson et al. to create the GM, WM and CSF probabilistic maps for each subject. Denoised and signal bias-corrected images were used in parallel to create a study-specific T$_{1w}$ template using modelbuild, an optimised pipeline using antsMultivariateTemplateConstruction2 that is an unbiased template building method developed in the ANTs package. Once both linear and nonlinear transformations were calculated for each animal, we compiled all the transformations calculated for each image and applied them once to the corrected images using antsApplyTransforms to limit interpolation effects. The resulting images were used to create TEBTA template by calculating the mean image for each normalised T$_{1w}$ sequence using Ginkgo. Probabilistic maps (GM, WM and CSF) were normalised using the previous linear and nonlinear transformations, and GM, WM and CSF priors were created by calculating the mean image for each normalised map using Ginkgo. Both templates and priors were used for spatial normalisation and segmentation for the VBM analysis (see the *Voxel-Based Morphometry Analysis* section). The complete procedure is described in the Supplementary Information.

**Creation of the TEBTA atlas.** The equine brain atlas provided by Johnson et al. was used as a starting point to create the TEBTA atlas. First, the Johnson's template was linearly and nonlinearly coregistered within TEBTA template using antsRegistrationSyNQuick. Then, both linear and nonlinear transformations were applied to each ROI of Johnson's atlas using antsApplyTransforms. Each normalised ROI was visually inspected to check boundaries and the accuracy of registration. Afterwards, some ROIs, such as *the corpus callosum, arbour vitae*, and ventricular systems, were updated/added using WM and CSF priors for a best fit of these ROIs to the TEBTA template. Eventually, additional subcortical ROIs, such as the septum, preoptic hypothalamic area, nucleus accumbens, and striatum, were drawn and delimited manually using fsleyes and itksnap. We implemented the methods previously used by Garin et al. for the establishment of a functional atlas in the mouse lemur to propose a valuable parcellation of the equine cortex[58]. Briefly, a multi-animal dictionary learning statistical analysis was performed with Nilearn (random_state = 0) on preprocessed rsfMR images (see the *Functional Imaging Analysis* section)[59]. A mask excluding the WM, CSF and subcortical areas was used to restrict the dictionary learning analysis to cortical functional data. A study based on 60 sparse components was selected for the final analysis. Each bilateral component was split into two unilateral regions and labelled left or right. This analysis led to the development of a 3D functional atlas composed of a mosaic of 55 local functional regions that were named using itksnap. The names of each ROI were defined using the names of the brain structures reported by Schmidt et al.[60], the AAL2 human brain atlas, and their structural connectivity (see the

*Creation of the TEBTA fibre atlas* section in the Supplementary Information and Supplementary Fig. 9).

**Creation of the TEBTA fibre atlas: structural connectivity of the equine brain.** Cortical ROIs resulting from dictionary learning segmentation of the equine cortex were used to identify the largest bundle tracts of white matter and aid in the identification of ROIs based on their structural connectivity. We constructed the fibre tractogram using the analytical Q-ball reconstruction model and streamline regularised deterministic (SRD) tractography algorithm available in Ginkgo[61]. From the obtained population tractogram, we selected the large bundles (length > 150 mm) using a two-step approach. First, bundles between each cortical ROI were selected by ROI-to-ROI selection, labelled and merged. Then, bundles between the thalamus and each cortical ROI were selected, as well as bundles between the cerebellum and each cortical ROI. With this selection, we expected to identify long cortico-cortical pathways (i.e. the *cingulum*) and thalamic projections, such as the somatosensory tract, to identify the somatosensory areas and the cerebellocortical tract to identify the motor areas. Non-relevant fibres previously selected within the bundle tract were filtered by length (>150 mm) and tortuosity (<2σ of the mean tortuosity). Second, the selected bundles were visually inspected, and 7 large bundle tracts (*cingulum*, corticospinal tract, anterior/posterior/inferior thalamic radiations, inferior longitudinal tracts and cerebellocortical tract) were identified based on morphology, position within the brain, location and by a comparison with fibre atlases available for humans and NHPs. Areas connected to these tracts were named according to their structural connectivity, position and the literature[60]. The complete procedure is described in the Supplementary Information.

**Voxel-based morphometry analysis.** For the VBM analysis, we used T$_{1w}$ images. The data were first denoised and signal bias corrected using Ginkgo and N4BiasFieldCorrection, respectively, and linearly coregistered to the TEBTA template using antsRegistrationSyNQuick. Coregistered data were then preprocessed with SPM. Each image was segmented into probability maps of GM, WM, and CSF using the default settings in the SPM8 toolbox and the TEBTA version of the GM, WM and CSF probability maps. The transformation matrices obtained were used to normalise the GM, WM and CSF probability maps obtained for each subject. The GM and WM probability maps of each scanning session were normalised to our stereotaxic space using the transformation matrices obtained here and resampled. Normalised GM and WM images were used for diffeomorphic anatomical registration using exponentiated lie algebra (DARTEL) to calculate diffeomorphic flow fields. Each normalised GM image was then warped using deformation parameters estimated with the DARTEL routine and then modulated to correct the volume changes that may have occurred during the deformation step. Finally, normalised, warped, and modulated GM images were spatially smoothed by convolving with a 4 mm full-width at half-maximum (FWHM) isotropic Gaussian kernel to create GMC maps. GMC maps were compared using an unpaired Student's t test in SPM to assess regional GM changes across the groups ('*with mother*' n = 11 versus '*without mother*' n = 12). A brain mask was used to constrain the analysis to the brain. For each cluster, the significance of the peak voxel was set as $p < 0.05$ (t score = 1.72; degrees of freedom = 21). The results are presented on an axial and sagittal brain slice series generated with Nilearn.

**Analysis of diffusion imaging data.** Because of its practical implementation, the NODDI model has become very popular for mapping tissue microstructures in vivo and ex vivo in both clinical[62,63] and preclinical applications[64]. The NODDI model relies on a biophysical model that separates the diffusion of water into three diffusive compartments (intraneurite, extraneurite and CSF)[28,62], which are assumed to be non-exchanging, contributing to global diffusion attenuation. The net

diffusion signal attenuation (*A*) corresponds to the following linear combination *A* resulting from a linear combination of the individual signal attenuations associated with each compartment: (1) the neurite compartment of water molecules trapped within axons and dendrites characterised by a volume fraction (*fic*), (2) the extraneurite compartment characterised by a volume fraction (*fec*) and (3) the CSF compartment containing free molecules with an isotropic displacement probability characterised by a volume fraction (*fiso*). Hence, the net signal diffusion signal *A* corresponds to the following linear combination:

$$A = fic \cdot a.ic + fec \cdot a.ec + fiso \cdot a.iso$$

We preprocessed the raw multishell diffusion imaging data and estimated each of the previously described fractions using Ginkgo. The multishell diffusion imaging data were corrected for magnetic susceptibility distribution (topup), motions and eddy current-induced distortions (eddy). The mean of the B0 data for each animal was calculated, and the resulting images were used to calculate a diffusion-weighted template using modelbuild. The resulting template was then normalised to the TEBTA template using antsRegistrationSyNQuick. Next, Ginkgo was used for the NODDI analysis (DwiMicrostructureField), and 4 maps were released: maps of FA, the intraneuritic fraction, the orientation dispersion index, and the isotropic fraction. Both linear and nonlinear transformations calculated using modelbuild and antsRegistrationSyNQuick were applied once to each contrast for spatial normalisation of the data within the TEBTA space using antsApplyTransforms. Then, the images were spatially smoothed through convolution with a 4 mm full-width at half-maximum isotropic Gaussian kernel. Each parameter was compared using an unpaired Student's t test in SPM to assess the changes in regional FA, ICF, ODI and isoF between the groups. A brain mask was used to constrain the analysis to the brain. For each cluster, the significance of the peak voxel was set as $p < 0.05$ (*t* score = 1.72; degrees of freedom = 21). The results are presented on an axial and sagittal brain slice series generated with Nilearn.

**Analysis of functional imaging data.** The EPI images were corrected for slice timing (slicetimer), motion (antsMotionCorr) and susceptibility distribution (topup). The mean of the EPI data for each animal was calculated, and the resulting images were used to construct a functional template using modelbuild. The resulting template was then normalised to the TEBTA template using antsRegistrationSyNQuick. Then, the images were detrended and low- and high-pass filtered (0.01 Hz–0.01 Hz). The effects of the six previously calculated motion parameters, including translations and rotations, WM, CSF and global signals were removed from the data through linear regression using Nilearn[24,25,50].

**Analysis of the fractional amplitude of low-frequency fluctuations.** The fractional amplitude of low-frequency fluctuation (0.01–0.08 Hz) values were computed using the previously processed 4D data. The fALFF function from the 1000 Functional Connectomes Project was used to determine the temporal and regional changes in GM occurring in the fALFF maps. Then, the fALFF maps were spatially normalised to the TEBTA template using both linear and nonlinear transformations calculated previously and applied once using antsApplyTransforms. Changes in regional fALFF changes between the groups were assessed using an unpaired Student's t test in SPM. A brain mask was used to constrain the analysis to the brain. For each cluster, the significance of the peak voxel was set as $p < 0.05$ (*t* score = 1.72; degrees of freedom = 21). The results are presented on an axial and sagittal brain slice series generated with Nilearn.

**Identification of the equine default mode network (DMN).** We identified the DMN in *Equidae* by calculating the mean Pearson correlation coefficient between time series of spontaneous variations in BOLD signals extracted from the functional ROIs defined as regions functionally homologous to regions involved in the DMN and DMN-like regions described in humans in numerous species (aG, aCC/pFC and RSC). Using the extracted data and a multiple regression analysis, we showed that spontaneous BOLD signal fluctuations in these regions were strongly correlated. For each cluster, the significance of the cluster was set as $p < 0.001$ (*t* score = 7.0544; two-tailed; degrees of freedom = 21; FDR corrected). The results are presented on an axial and sagittal brain slice series generated with Nilearn.

**Behavioural observations**
Behavioural observations occurred during the day between 10:00 am and 3:30 pm. For each observation period (+1 month, +3 months, and +7 months after maternal separation), foals were observed across the same week for 2–5 days. In each period, foals were observed for 10 to 20 h in total. Behavioural observations consisted of scan sampling[65]. Every 15 min, the activity of each foal, whether it was isolated or near another animal (≤10 m), the identity of its closest neighbour when relevant and the distance between them in metres were recorded. A total of 40 to 84 scans/period/foal were therefore analysed. The proportion of scans recorded in each of these activity categories was calculated to estimate the time the foals spent feeding (drinking and suckling included), exploring the environment, resting (standing or lying down), interacting socially in any form (agonistic, affiliative, or otherwise), and interacting positively and actively (allogrooming or playing socially). In addition, a centrality score, or eigenvector, was calculated based on the spatial proximity with the closest neighbour[66]. Two individuals were considered close when the interindividual distance was less than 1 m (i.e. a distance enabling the individuals to interact without having to move). The eigenvector was obtained using SOCPROG 2.9[66] with a two-entry proximity matrix. The eigenvector represents the total number of connections an individual has, including the connections between its partners. The higher this index is, the more central the individual is within their social group. Finally, an 'isolation' score was calculated based on the proportion of scans recorded when the foal was isolated (+10 m away from another groupmate). The more time the foal spent isolated, the less gregarious and the more socially independent the foal is. All these parameters were calculated for each foal for each period of observation (+1 month, +3 months and +7 months after maternal separation).

**Behavioural tests**
Apart from the social isolation test, all tests were conducted in the foals' home environment. Each test was performed on a separate day within the same week of the 7th month post-maternal separation period. On the morning of each test day, the entire group (i.e. mothers, '*with mother*' foals, and '*without mother*' foals) was assembled and confined in half of the collective stable using removable tubular barriers and gates. This arrangement ensured that the test area was adjacent to the group area and familiar to the foals, thereby minimising any potential novelty effect or social isolation. Each individual test started in a standardised manner. A fully opaque wooden panel (dimensions: height 2 m, length 4 m) was installed to create a specific corner in the test area that remained hidden from the rest of the group, ensuring that the stimulus (such as an unfamiliar horse or object) was visible and accessible only to foals undergoing testing at that moment. Two experimenters entered the group area and approached the animal selected for testing, with the order of selection being semirandom and varying for each test session. The foal was then fitted with a head collar, with one experimenter holding a lunge line and the other experimenter walking behind the foal and guiding it with their arms open. Together, they led the foal calmly through the gate separating the two sections to the test area. Upon reaching the test area, the foal was released approximately 5 m past the gate and positioned at a

reasonable distance (20 m) from the designated 'stimulus' area. This setup enabled the test to begin, allowing the foal the freedom to approach the stimulus or remain away from it (see Supplementary Fig. 1 for an illustration).

**Sociability test.** Before the beginning of the test, an unfamiliar horse (a female Welsh pony, 23 years old) was brought to the test area. Selected for its calmness and lack of reactivity to social isolation, the unfamiliar horse was tied up using its head collar and provided with forage and a bucket of water to ensure that it remained relatively motionless and positioned parallel to the test area. The panel separating the unfamiliar horse and the test foals consisted of a feed-through panel (i.e. an ensemble of vertical metallic bars from the floor to the top, allowing the foal to pass the head and neck through the panel but nothing more). With this arrangement, the foal was allowed to approach and enter in physical contact with the unfamiliar horse using its head only while still allowing the unfamiliar horse to step away if desired. Throughout the procedure, the unfamiliar horse remained calm and accessible to all test foals. Interactions between animals were mainly neutral, characterised by reciprocal exploration, or no interaction occurred at all. The test duration was set to 10 min, starting with the release of the test foal from its initial position. The latency before the tested foal contacted the unfamiliar horse was recorded, with a maximum score of 601 s assigned if contact did not occur before the end of the test.

**Fearfulness test: novel object test (individually).** Before testing, an unfamiliar object (a pink fitness ball, diameter: 65 cm; see Supplementary Fig. 1) was positioned in the test area away from the view of the group. The tested foal was then brought to the test area using the same procedure described above and released into the area for a duration of 10 min. During this time, the latency before the foal contacted the unfamiliar object was recorded. A maximum score of 601 s was assigned if contact did not occur before the end of the test.

**Fearfulness test: novel surface test (individually).** Here, an unfamiliar plastic tarp (dim: 3 m × 3 m) with a bucket containing pellets placed on top (bucket positioned centrally) was positioned before testing out of sight of the group (see Supplementary Fig. 1 for an illustration) as previously described[67]. For this test, the test foal was led 5 m from the surface by one experimenter. Once in place, a second experimenter held the bucket and brought it to the foal, allowing it to ingest a mouthful of pellets. Then, the second experimenter walked back to the surface, ensuring that the foal remained attentive, and then returned the bucket to the centre. Afterwards, the foal was released by the first experimenter. With this setup, the foal was required to approach the tarp and place its two forelegs on it to access and consume the pellets. The test concluded either when the foal completed this action or after 6 min if the foal did not step on the surface at all. The latency before the foal stepped on the surface for the first time was recorded. A maximum score of 361 s was assigned if the foal did not step on the surface before the test concluded.

**Fearfulness test in the social context: novel object recognition test in groups.** The novel object was a foldable tunnel covered with plastic fabric (diameter: 0.6 m; height: 4), which is typically used for agility practice in dogs. Two experimenters led all 11 or 12 foals from the same social group to the test area simultaneously, while the mothers remained in the initial group area. Upon the arrival of all the foals in the test area, the novel object was unfolded and immediately suspended 80 cm above the floor using a sliding string connected to the ceiling of the building. This string was moved by a third experimenter, ensuring that the object was visible to all animals present at the same moment. Once the object was in place (see Supplementary Fig. 1 for an illustration), all the animals were allowed to move and explore freely for a period of 30 min. During this time, the latency before each foal contacted the unfamiliar object was recorded. A maximum score of 1801 s was assigned if contact did not occur before the end of the test.

**Social isolation test.** This test, performed during the 4th month after maternal separation, was adapted from previous studies[68]. Here, each foal was individually led to a separate building located 150 m away from their home stable using a head collar and a rope, with one experimenter leading the foal and another providing guidance from behind. After arriving, each foal was promptly placed in an individual stable (4.5 m × 3.5 m) with the lower part of the door closed for 90 s. From this position, the foal could see the experimenters but not any foals. It could hear sounds from foals that were not included in our study. The number of vocalisations (neighing) by each foal during the test was recorded.

**Standardised training and observation of care compliance in foals.** Over a period of four consecutive weeks during the 5th month after treatment, we standardised and observed the routine hoof care training that all foals in the experimental unit underwent for health purposes. The training aimed to familiarise the foals with being touched on their limbs and voluntarily lifting their hooves using a cooperation-based approach with food rewards. Hoof handling training was performed to ensure that the animals would tolerate routine farriery. We recorded the number of sessions required for each foal to meet the predefined criteria for hoof lifting acceptance as a proxy to assess the duration of training needed to achieve compliance. During the final week, we observed the farriery session, which was a necessary procedure for all animals. During this session, we also recorded the number of attempts each foal made to withdraw its hoof under standardised conditions.

### Physiological assessment

Blood samples were collected from all foals between 10:00 AM and 12:00 PM to avoid circadian variation, after which each animal was weighed immediately under constant conditions and without restraint. Sampling occurred once before the maternal separation treatment and then at +1, +3 and +7 months after separation. All samples were collected in the foals' home collective stable, where the animals were handled calmly without the need for additional equipment. The foals were gently guided through a passage formed by a system of metal barriers leading to a scale to facilitate blood collection and weighing. A small front gate was momentarily closed to stop the foal from briefly being on the scale. Blood samples were drawn swiftly during this short pause—lasting only a few seconds—by an experienced handler standing alongside the foal. The gate was immediately opened after sampling. This setup allowed us to safely position the animal without halters or physical immobilisation.

On each sampling day, 20 mL of blood was collected from each foal and then centrifuged to extract the plasma, which was stored at −20 °C until analysis. At the end of the experiment, plasma concentrations of cortisol, oxytocin, glycaemia, cholesterol, IGF-1 and insulin were measured to capture baseline levels of these physiological markers (see the Supplementary Information for technical details on the procedure, and Supplementary Table 3 and Supplementary Fig. 10 for comparisons of pretreatment concentrations showing no initial differences between groups).

### Data analysis

All the statistical analyses were performed with R (version 4.2.2)[69]. Two foals had missing values: one for long-term behavioural tests and the other for MRI data. The missing data were imputed using MFA, as implemented by the imputeMFA function within the missMDA package[70]. Additionally, given that the foals were reared in same-sex

groups, any potential effect of sex was mitigated using the ComBat method from the sva package[71]. This method enables adjustment for batch effects in datasets where the batch covariate, in this case, sex, is known, employing an empirical Bayesian framework. MFA is performed on all the parameters of the imputed and corrected data using the FactoMineR R package[72,73] to explore differences between foals housed with or without their mothers. Unlike principal component analysis, MFA organises variables into groups, each carrying equal weight in the analysis. In this study, the six defined groups included *weight gain*, *non-social behaviour*, *non-social personality traits*, *sociality*, *MRI* and *physiology*. The correlation structure among the data groups was subsequently investigated using the mixOmics R package employing the DIABLO framework (Data Integration Analysis for Biomarker discovery using Latent cOmponents)[74,75]. This comprehensive multivariate approach optimises correlations across various data types and conducts discriminant analysis to uncover a specific signature for the foal groups. The underlying method used is a multiblock projection to latent structures (MP–PLS) model with sparse discriminant analysis. PLS components, which are linear combinations of variables, were constructed to maximise the sum of covariances across all blocks (i.e. groups of data). The function perf() was run with 3-fold cross-validation repeated fifty times to choose the number of components. An examination of the performance plot revealed increases in both the overall and balanced error rates (BERs) as the number of components increases from 1 to 2. All distance measures appeared to yield similar accuracy, leading us to opt for the centroid distance. Based on this distance metric and the BER, we set the number of components to 1. Furthermore, the weights of all pairwise covariances were determined using the design matrix. In this study, we sought to determine the links between the variables of the different blocks through this analysis. Consequently, we opted to establish the design matrix value at 0.8 for all block pairings, facilitating a thorough exploration of these associations. By employing the sparse option alongside discriminant analysis, the most discriminant variables between foal treatments were selected from each group of data. We chose the optimal number of variables to select in each data group using the tune.block.splsda() function for a grid of keepX values for each type of data. This function was set to favour a relatively small signature while allowing us to obtain a sufficient number of variables for downstream interpretation. The function tune.block.splsda() was run with 3-fold cross-validation and repeated fifty times. As this function is time-consuming, the grid of values tested for each data group was reduced to a smaller number of repetitions. Individual tests were subsequently conducted on the kinetic variables identified through the preceding multivariate analysis. Aligned ranks transformation ANOVAs were performed to examine the interaction effect between treatment and time while also incorporating a random effect of individuals into the model[76]. Appropriate post hoc comparisons were conducted to assess the differences in treatment at each time point, as well as the temporal evolution of each measurement[77]. These analyses were performed using the R package ARTool. The same tests were applied to the area under the curve of each kinetic parameter to obtain an overall difference. All adjusted p values less than 0.05 were considered significant.

### Reporting summary
Further information on research design is available in the Nature Portfolio Reporting Summary linked to this article.

## Data availability
The whole dataset, along with the detailed statistical analyses, have been deposited in Zenodo (https://doi.org/10.5281/zenodo.13969827). The Turone Equine Brain templates and atlas toolkit have also been deposited in Zenodo and are publicly available at https://doi.org/10.5281/zenodo.10731031. Source data are provided with this paper.

## Code availability
The codes used for the magnetic resonance imaging (functional, voxel-based morphometry and diffusion imaging) analysis have been deposited in GitHub (https://github.com/DavidBarriere/Equisobrain). The codes used for statistical analysis have been deposited in Zenodo (https://zenodo.org/records/13969827).

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

## Acknowledgements
M.V. and M.K. gratefully acknowledge support from the European HORIZON 2020 Marie Skłodowska-Curie Actions (MSCA) (project number: 101033271, MSCA European Individual Fellowship, https://doi.org/10.3030/101033271, fellow: MV, host supervisor: MK) and the *Institut Français du Cheval et de l'Equitation* (IFCE) for co-funding this project (DEVELOPPEMENTPOULAIN project, M.V., M.K.). This work was also supported by the French National Research Institute for Agriculture, Food and the Environment (INRAE), which, through its specific experimental units, Unité Expérimentale de Physiologie Animale de l'Orfrasière and the PIXANIM imaging platform belonging to the UMR Physiologie de la Reproduction et des Comportements, provided both animals and technical facilities for successful MR imaging on domestic horses. This work also benefited from the Phenotyping–Endocrinology Laboratory for the hormonal assays and the IT infrastructure of the ISLANDe platform, particularly a computing cluster financed by the European Regional Development Fund n° 159037. The authors greatly appreciate the entire 'equine' team of the Animal Experimental Unit for their technical support and consistent commitment throughout data collection and animal care, especially Thierry Blard, Melinda Rousseau, and Tiphaine Aguirre-Lavin. The authors are deeply grateful to the whole team of the Veterinary Clinic of the Nouvetière (Sonzay, France). Finally, the authors thank B. Piégu for computer system administration and for help with ANT-cluster deployment and L. Croizier for her valuable assistance with behavioural data collection.

## Author contributions
Conceptualisation: M.V., F.R., L.C., L.L., M.K., and D.A.B. Methodology development: M.V., F.R., H.A., F.E., I.L., C.D., C.P., I.U., and D.A.B. Data collection: M.V., F.R., A.G., H.A., F.E., J.D., A.L.L., P.B., Y.G., and D.A.B. Project administration: M.V. and M.K. Data validation and analysis: M.V., G.L., and D.A.B. Writing of the manuscript: M.V., G.L., and D.A.B. Commenting on and editing the manuscript: H.D., J.D., L.C., L.L., A.L.L., H.A., and M.K.

## Competing interests
The authors declare no competing interests.
