## [Transparent Peer Review file · Nature Communications]

Affiliative behaviours regulate allostasis development and shape biobehavioural trajectories in horses

Corresponding Author: Dr David Barrière

Version 0:

Reviewer comments:

Reviewer #1

(Remarks to the Author)

The work is truly outstanding in its attempts to combine behavioural, physiological, affective states and brain imaging data. There are minor English corrections needed in the text (line 106 for example).

What are the noteworthy results?

The authors present strong evidences of the positive impact of the presence of mares, together with their foals for longer than 6 months of age, when contrasting with foals kept in the same group, but separated from their mares at 6 months of age. The results reported are present in brain organization measured using MRI, gray matter in areas of relevance to cognition, emotional regulation, feed intake (and other functions). Basal levels of cortisol appear to be lower, three months after maternal separation in the foals kept with their mares. The behavioural data, obtained using scans every 15 minutes, provide a rather confusing scenario as the foals without their mothers, suckled more than the foals with their mothers what is very unlikely. Data on social behaviour using scan sampling can yield very unreliable results, especially with long intervals as the 15 minutes used in the present experiment.

Will the work be of significance to the field and related fields? Absolutely, the study is unique in bringing together so many different measures in the responses of foals to maternal separation. Mare and foal interactions are fascinating and the complex nature of the design make the paper outstanding.

How does it compare to the established literature? It is one of the most complete publications that I have encountered in this area.

If the work is not original, please provide relevant references.

The work is truly original.

Does the work support the conclusions and claims, or is additional evidence needed?

I consider that the enthusiasm of the authors for the findings did not let them consider an alternative hypothesis to explain the differences that they encountered.

Are there any flaws in the data analysis, interpretation and conclusions? Do these prohibit publication or require revision?

I do not consider it to be "fatal flaws", however it is needed to offer alternative explanations to the reported findings. The decision on maintaining the 6 weaned foals, either males or females, together with the foals with their mares, created a rather different social dynamics in the group. It is possible that a social network, with the pre-existing bond between mare and foals, not only gave advantage to the non-weaned groups, but provided an unbalanced social environment for the weaned foals, where they negotiated a potentially more hostile environment. The paper will benefit from more detailed behavioural observations, using focal sampling and continuous recording, where data on social interactions could unravel the social interactions among the foals. What I hypothesize is that the presence of the foals and their mothers, together with the foals without mothers, compromised the weaned foal's social behaviour, in a very significant way. The foals with their mothers would, in theory, have a very strong social network (bonds), in contrast with the weaned foals, which will be lacking maternal protection, in an environment where the other group would have maternal protection. This situation could exacerbate the differences among groups, making it very good for the foals with their mother but exaggerating the challenges in the weaning group. This could have been corrected adding another control group with weaned foals kept with unrelated mares, removing the impact of social network (previous bonds) on the behaviour of the foals. The study needs better behavioural data, to characterize social interactions in the group in a more detailed way.

The MRI data needs to be better discussed in the areas involved in the contrast between the weaned and non-weaned groups, to offer an understandable explanation. Amygdala and its role in fear responses and empathy is not explored. The hippocampus, as the hub for memory, cognitive processes and emotions is also not discussed appropriately. The thalamus

and its role in controlling pain, could be mentioned. The relationship between the hypothalamus and endocrine regulation, more than weight gain, is also relevant to the work.

dology sound? Does the work meet the expected standards in your field?

Yes, I do consider the methodology sound and reflect the acceptable standards in our field of research,

Is there enough detail provided in the methods for the work to be reproduced?

More details are needed in the behavioural data. Blood collection is mentioned to have occurred without restraint of the animals, is that correct?

Conclusion

In summary, this is an outstanding manuscript which needs some improvements to be considered for publication, particularly related to the behavioural data collection. I would like also to read an alternative hypothesis mentioned in the discussion acknowledging the possibilities that the experimental design may have exacerbated the negative responses of the weaned foals.

Reviewer #2

(Remarks to the Author)

The study by Valenchon et al. used MRI measurements (anatomical, functional, and diffusion), along with physiological and behavioral measurements, to investigate the effects of caregiver loss on brain parameters and individual behavior. The study is very well written and represents a milestone in the field of social neuroscience.

A key strength of the study is that the authors have carefully considered all the dependent variables in terms of MRI, physiology, and behavior. By combining multimodal imaging, they have created an atlas that will be essential for future MRI studies of the hours, which could make this work highly influential. These strengths demonstrate the study's importance to the field and make it a valuable contribution to publish here.

However, despite these strengths, I have identified some issues that need clarification before the manuscript can be considered for publication. My primary concern is the timing of the MRI imaging and the absence of a baseline measurement prior to separation. Additionally, fMRI is an indirect measure of brain activity, which can be influenced by physiological changes, potentially leading to a misinterpretation of brain network changes as functional plasticity. The authors should revisit the results and carefully address this issue to avoid such misinterpretation.

I have listed my comments below and would be happy to see an improved version of the manuscript after revisions.

Major Comments

1. Baseline Measurements:

o In the study, sampling and monitoring began one month after separation. This results in a missing baseline measurement. Why did you not record most of the measurements before separation to establish a baseline for comparison?

2. Timing of MRI Measurements:

o Why were MRI measurements conducted only one month after separation? If one month is sufficient to observe the effects of caregiver loss, why were other measures recorded at 3, 5, and 7 months?

3. Lack of Controls and Quality Assessments: Since this study is a pioneering effort in the field, it lacks several necessary control measures and quality assessments.

- You used Spin-Echo EPI for resting-state fMRI. However, in both animal and human research, Gradient-Echo EPI (GE-EPI) is the standard sequence. Could you clarify why you chose Spin-Echo EPI instead? Please add a supplementary figure showing raw images for quality assessment.

- It is essential to examine FA maps in subject space before running comparisons. Please provide these maps along with the FA value range.

- For all MRI data (DWI and fMRI), please report SNR (Signal-to-Noise Ratio) and tSNR (temporal SNR) where applicable.

- It is crucial to report motion parameters for resting-state fMRI, as motion can easily introduce artificial correlations.

- For resting-state fMRI, you used a ~4-second repetition time (TR), which is quite long. Such a low sampling rate may lead to aliasing effects, where respiration-related fluctuations modulate resting-state fMRI signals. Since caregiver loss affects physiological parameters, the observed changes in resting-state networks could be driven by physiological variations rather than genuine functional plasticity of brain regions. This needs a clear explanation. If respiration data were recorded, it would be valuable to report the respiration rate for both groups during MRI scans. Additionally, as shown in this study (<https://pmc.ncbi.nlm.nih.gov/articles/PMC9645809/>), respiration can introduce artifacts in rs-fMRI and is also associated with a respiration-related brain network. This factor should be considered when interpreting the results.

4. Statistical Tests and Reporting Issues:

- The statistical tests used in the study are unclear. You frequently mention Student's t-test, but since you are comparing two groups, paired t-tests would be more appropriate. Were these tests one-tailed or two-tailed? This needs to be explicitly stated in the text.

- You applied FDR correction for multiple comparisons, but FDR only provides a corrected p-value. What cluster size threshold was used to define significant clusters? This information is missing.

- In Figure 3, I see small blue clusters (e.g., first image, front; middle image, top row) and yellow clusters (present in all figures except the first top left). However, these are not mentioned in the results. Please clarify and update the text accordingly.

- Another surprising issue is that many reported t-test values are identical ($t = 1.7207$) for different comparisons (line 126 for GMC, line 141 for FA, and line 144 for OD). This seems highly unlikely. Please double-check your test results.

- The value $t = 1.7207$ is quite low and corresponds to the critical value of a one-tailed t-test. Please clarify whether this was intentional.

- Wherever possible, report the actual p-values instead of just stating significance thresholds.

5. Issues in Figure 4 Reporting:

- In Figure 4a, a yellow cluster suggests a significant effect during maternal presence, but this is not reported in the results.

Why?

- Similarly, in Figure 4b, a blue cluster is visible but not discussed. These findings should be addressed in the text.

6. Default Mode Network Identification:

- How did you define the default mode network (DMN)?

- What was the correlation value between these regions?

- Were there additional regions correlated with this network that were not reported?

- Why was $p < 0.001$ used as a significance threshold instead of $p < 0.05$?

7. Multiple Regression Analysis and Default Mode Network (DMN) Interpretation:

- The multiple regression analysis used to identify the DMN is not well explained. Please provide a detailed description of the methodology.

- How did you detect anticorrelation between somatosensory regions? More details are needed to support this claim.

- Was this network also visible in ICA (Independent Component Analysis)? If so, please clarify.

8. Figure 8 and MRI Metrics:

- What MRI-derived metrics were used to plot the matrix in Figure 8?

- Since MRI was recorded only one month after separation, but many behavioral and physiological measures were recorded much later (e.g., behavioral tests at 7 months), there is a significant time gap. Given the aging effects on MRI-derived parameters, comparing these results with physiological or behavioral measures recorded much later is problematic. This limitation needs to be discussed in detail.

9. Misinterpretation of fMRI Findings in the Discussion:

- The statement in the discussion:

“Providing a better understanding of how structural and functional brain organizations are linked with physiology and/or behavior”
is misleading.

- fMRI measures BOLD signals, which depend on blood oxygenation levels. Physiological changes can easily influence these signals, meaning that observed correlations between physiology and BOLD signals should not be directly interpreted as functional plasticity. The results do not necessarily indicate brain function is changing but rather that physiological variations may be driving the observed effects.

- A more physiology-independent technique is needed to validate these findings.

- Additionally, the papers cited in support of this claim focus on disease symptoms and cognition, not physiological changes and brain networks. Please clarify this distinction.

10. Discussion Needs Revision Regarding Physiological Effects on BOLD fMRI:

- The discussion needs to acknowledge that BOLD signal variations may be due to physiological changes rather than true functional brain network changes due to the mother separation.

- Please refer to the following studies, which discuss the impact of physiology on BOLD signals:

<https://www.sciencedirect.com/science/article/pii/S0006899314016916>

<https://www.mdpi.com/2072-6643/13/7/2134>

- The interpretation of fMRI results should be revised accordingly.

Minor Comments

1- Figure Quality:

The quality of the figures is too low, making the numbers and labels unreadable. Please improve the resolution to ensure readability.

2- Figure 1g:

The label for Figure 1g is missing. Please ensure it is clearly labeled on the figure.

3- Figure 1d Clarification:

In Figure 1d, it is unclear whether the test was done only once after 7 months of maternal separation, or if it was repeated at 1 month, 3 months, and 5 months as well. Please clarify and include this information in Figure 1a.

4- Line 100 - “Post-treatment”:

In line 100, the term “post-treatment” is used. Could you clarify what this refers to?

5- Line 105 - Summary of Variables:

In line 105, you mention “a total of 76 variables.” Could you provide a summary of the variable names here to clarify for readers which variables were included?

6- Multiple Factor Analysis (MFA):

What was included in the MRI imaging for the Multiple Factor Analysis (MFA)?

The other variables are quantified, but how were these MRI variables quantified? Please clarify this process.

It would be helpful to add a supplementary table listing all the variables used for this analysis for better clarity.

7- Figure 1 – Dim 1 and Dim 2:

In Figure 1, could you explain what Dim 1 and Dim 2 represent?

8- Figure 1g Clarity:

The labeling in Figure 1g is unclear.

What exactly is plotted here?

What do the light and dark colors represent?

Why do some colors appear before time point 2?

What do the numbers on the horizontal axis mean?

The figure suggests that MRI imaging is the least significant variable for separating the two groups. Could you discuss how the results would change if MRI measurements were excluded from the analysis?

9- Figure 1g – Text Readability:

The text in Figure 1g is not readable. Please adjust the font size or improve the clarity of the text.

10- Line 115 - “Specific Brain Metrics”:

In line 115, you refer to “specific brain metrics.” Could you clarify which metrics are included here? Please provide more detail about the brain metrics used in the analysis.

11- Figure 3 – Maximum Values:

In Figure 3, what are the maximum values reported? Please add color bars with the corresponding maximum values visible in the figure to improve clarity.

12- Line 157 – Frequency Band Selection:

Why did you select the frequency band of 0.01–0.08 Hz? Please explain the reasoning behind this choice.

13- Figures 6 and 7 – AUC Clarification:

In Figures 6 and 7, could you clarify what is shown in the left plot of all subplots?

Additionally, could you explain what AUC stands for and how it is being interpreted in this context?

Reviewer #3

(Remarks to the Author)

Review of “Affiliative behaviours regulate allostasis development and shape biobehavioural trajectories in horse”.

This is an interesting study evaluating the impact of mother-separation in 6-month-old colts on short-term (6 month) measures of social behavior, physiology, and reactivity in a variety of behavioral novelty tests. The authors show that maternal removal from the herd (and hence maternal separation) also alters grey matter concentrations in areas relevant for emotional processing (e.g., amygdala, hippocampus, and hypothalamus) and the strength of Default Mode Networks, relative to mother-present colts.

Overall the results are explained appropriately. However, the links between maternal separation and the outcome measures remain correlational in nature, and language in the title and the discussion of the results should replace causal language (“affiliative behaviors regulate ... and shape biobehavioral trajectories” is too causal). The analyses show that the variables are correlated, but not necessarily causally related. As a consequence of this, the Discussion should also include a more detailed evaluation of alternative hypotheses that could account for the correlational differences between the two groups.

The data for the physiological measures (e.g., cortisol, cholesterol, triglycerides) reveal some interesting differences between the treatment groups. However, the authors do not report whether the two treatment groups were statistically equivalent in these measures prior to maternal separation. The Methods section indicates that a blood sample was collected just prior to maternal separation, so the samples are apparently available to analyze those data, but the paper only reports these measures (and the behavioral measures, as well) for the sampling periods AFTER maternal separation. The Introduction establishes ‘allostasis’ as an important organizing construct for this experiment, but the Discussion does not provide a sufficiently thoughtful discussion of how the data in the present experiment advise the concept that allostatic responsiveness is affected by maternal separation.

The timeline for assessing the impact of mother-removal is very limited, given the longevity of horses. There are no data on the long term impact of this manipulation that would be necessary to conclude that maternal separation had an impact on long-term behavioral/neural/physiological) trajectories, beyond a short-term impact during the 6 months separation period studied in this paper.

Minor points/queries:

- The paper does not describe (or really even discuss) the significance of the Area Under the Curve measures reported for Behavior and Physiological measures. What additional information is made available by these measures above and beyond what is reported for mean values (rates of behavior or concentrations of analytes)?
- As far as this reviewer can determine, this manuscript presents the first data on MRI and fMRI data from horses from this laboratory. The description of the Methods for these measures, and the analyses performed on the data, are appropriate but lengthen (and potentially detract from) the main argument regarding the impact of maternal separation on behavioral, neural, and physiological function.
- The paper includes subjects of both sexes, and presumably included ‘sex’ as a factor in the analyses. However, the paper does not describe whether no sex differences were detected or whether some of the changes in measures as a consequence of maternal separation differed between males and females.
- The measure of social proximity (≤ 10 m from a potential social partner) seems like a liberal measure of social proximity. Some indication of the size of the paddock may help clarify this point.

Version 1:

Reviewer comments:

Reviewer #1

(Remarks to the Author)

The authors did address the comments that I issued in the first review in an appropriated manner.

General comments

What are the noteworthy results?

Absolutely yes.

Will the work be of significance to the field and related fields?

I have stated previously that the answer is positive, as the integrated study on the consequences of maternal care on brain, behaviour and physiology in horses offer unique opportunities for comparative studies.

How does it compare to the established literature?

The approach is very disruptive as it brings together measures on brain, behaviour and physiology.

If the work is not original, please provide relevant references.

Very original.

Does the work support the conclusions and claims, or is additional evidence needed?

New evidences were added in the revised version, specially regarding behavioral observations.

Are there any flaws in the data analysis, interpretation and conclusions? Do these prohibit publication or require revision?

My concerns regarding the weaned group social stress, as a result of the presence of mothers of the unweaned group were addressed in the text, offering alternative explanations to justify the findings.

Is the methodology sound? Does the work meet the expected standards in your field?

Yes, the methodology is acceptable. The pitfalls in the behavioral data collection were addressed in the revised version,

Is there enough detail provided in the methods for the work to be reproduced?

New information on the methodology was added to the manuscript.

Reviewer #2

(Remarks to the Author)

The submitted revision is significantly improved, and I truly appreciate the hard work of the authors. The manuscript is getting very close to its final form.

There is only one minor issue that could make it difficult for readers to fully understand the results.

In the figures, the activation maps also show clusters that did not survive correction for multiple comparisons. Why are these non-significant clusters included? Why not present only the clusters that passed the multiple comparison correction?

At the very least, the figure captions should clearly indicate which clusters are significant and which are not. As it stands, it is difficult to interpret the true results from the figures.

I strongly suggest presenting only the clusters that survive the multiple comparison test, to ensure clarity and avoid confusion.

We sincerely thank all the reviewers for their positive feedback, which has been incredibly encouraging and helpful in refining our manuscript. Their work has been invaluable in improving the previous version, and we hope to have addressed all the points raised adequately."

Reviewers' comments

Reviewer #1 (Remarks to the Author):

The work is truly outstanding in its attempts to combine behavioural, physiological, affective states and brain imaging data. There are minor English corrections needed in the text (line 106 for example).

We sincerely thank the reviewer for the positive feedback. We have made efforts to improve the clarity and accuracy of the language throughout the revised version.

Lines. 105-107: we have revised the wording as follows: *“At the end of the experiment, a total of 76 variables (see supplementary material) were collected for 23 animals, as one female belonging to the “with mother” group was removed from the experiment for medical reasons.”*

What are the noteworthy results?

The authors present strong evidences of the positive impact of the presence of mares, together with their foals for longer than 6 months of age, when contrasting with foals kept in the same group, but separated from their mares at 6 months of age. The results reported are present in brain organization measured using MRI, gray matter in areas of relevance to cognition, emotional regulation, feed intake (and other functions). Basal levels of cortisol appear to be lower, three months after maternal separation in the foals kept with their mares. The behavioural data, obtained using scans every 15 minutes, provide a rather confusing scenario as the foals without their mothers, suckled more than the foals with their mothers what is very unlikely. Data on social behaviour using scan sampling can yield very unreliable results, especially with long intervals as the 15 minutes used in the present experiment.

We sincerely thank the reviewer for his/her detailed feedback and for acknowledging the strengths of our study.

Regarding suckling behaviour: We found that initial formulation may have led to a misunderstanding regarding suckling behaviour. To clarify, the foals in the ‘without mother’ group did not suckle at all, as expected. Only foals in the ‘with mother’ group engaged in suckling, though the total time spent suckling was relatively short. To ensure a fair comparison of overall feeding time, we included suckling within the total time spent feeding (expressed as a percentage of scans). This allowed us to avoid underestimating the feeding time in foals with their mothers. Despite this adjustment, foals in the ‘with mother’ group still spent less time feeding than those without their mothers, and yet they gained more weight. This led us to hypothesize a higher feeding efficiency in foals with prolonged maternal presence, which we

la science pour la vie, l’humain, la terre

discuss in the manuscript in relation to potential explanatory factors, including the role of lactation. We revised the text of the manuscript to make all these clearer.

Revised text (Lines 192-202) : *“Analyses of the percentage of scans spent in each behavioural activity combining all periods of observations (from the 1st to the 7th month of prolonged maternal presence) using Aligned Ranks Transformation ANOVAs showed that animals belonging to the “maternal presence” group displayed higher levels of exploration ($p = 0.002$), rest ($p = 0.02$), social interaction (for any type: $p < 0.001$; and for affiliative ones: $p < 0.001$) and lower time spent feeding ($p < 0.001$) (Fig. 6a). To ensure a relevant comparison between experimental groups, suckling was included in the total feeding time. For the “with mothers” foals, the percentage of scans showing suckling was $2.08 \pm 0.36\%$ at 1-month post-weaning, $2.19 \pm 0.55\%$ at 3 months post-weaning, and $2.83 \pm 0.70\%$ at 7 months post-weaning (mean \pm SE). No foals belonging to the “maternal absence” group were observed suckling. Despite accounting for suckling in the calculation of total feeding time, “with mothers” foals still spent significantly less time feeding overall compared to “without mothers” foals.”*

Regarding the scan sampling method, we acknowledge the potential limitations of scan sampling with 15-min intervals. In our study, we selected this interval to balance observer feasibility with the need to capture behavioural parameters over extended periods. The 15-min interval choice ensured that consecutive scans remained sufficiently independent from one another, as animals had enough time to change activities between scans. In other words, we choose to prioritize a longer observation window over multiple days rather than using shorter intervals within a more condensed period. Furthermore, the repeated scan sampling across multiple time periods and on the same individuals increased the statistical power of our analyses. This approach allowed us to extract valuable information not only on individual behaviours, including social interactions, but also on spatial proximities, which were used to compute social network metrics such as centrality. We acknowledge that the initial version of the manuscript may not have sufficiently highlighted the breadth and richness of our behavioural dataset, which includes both individual activities and social/spatial parameters. Given the level of details recorded in our observations, that were also completed by a series of behavioural tests, we believe that scan sampling remains highly informative and appropriate for the objectives of our study. However, we fully agree that future work would benefit from additional in-depth observations, such as focal sampling, to further refine social analyses.

Revised text (Lines 382-390): *“Nevertheless, an alternative hypothesis could posit that the presence of the mother might have provided indirect benefits through pre-existing affiliative bonds within the group. These additional social resources may have further amplified the contrast with weaned individuals, increasing the level of social challenge for weaned horses compared to foals benefiting from the presence of their mothers. Future studies using focal sampling or continuous recording to capture overall social interactions will be necessary to unmask the affiliative dynamics and disentangle the respective contributions of maternal presence and group-level affiliative structure to better understand how prolonged maternal presence shapes developmental trajectories.”*

Will the work be of significance to the field and related fields?

Absolutely, the study is unique in bringing together so many different measures in the responses of foals to maternal separation. Mare and foal interactions are fascinating and the complex nature of the design make the paper outstanding.

la science pour la vie, l'humain, la terre

UMR (* nom de l'UMR)

Adresse

Code postal et ville

Tél. : 00 00 00 00 00

Rejoignez-nous sur :

Site internet de l'UMR

INRAE

How does it compare to the established literature?

It is one of the most complete publications that I have encountered in this area.

If the work is not original, please provide relevant references.

The work is truly original.

We would like to thank the reviewer once again for his/her strong support for our work.

Does the work support the conclusions and claims, or is additional evidence needed?

I consider that the enthusiasm of the authors for the findings did not let them consider an alternative hypothesis to explain the differences that they encountered.

Are there any flaws in the data analysis, interpretation and conclusions? Do these prohibit publication or require revision?

I do not consider it to be “fatal flaws”, however it is needed to offer alternative explanations to the reported findings. The decision on maintaining the 6 weaned foals, either males or females, together with the foals with their mares, created a rather different social dynamics in the group. It is possible that a social network, with the pre-existing bond between mare and foals, not only gave advantage to the non-weaned groups, but provided an unbalanced social environment for the weaned foals, where they negotiated a potentially more hostile environment. The paper will benefit from more detailed behavioural observations, using focal sampling and continuous recording, where data on social interactions could unravel the social interactions among the foals. What I hypothesize is that the presence of the foals and their mothers, together with the foals without mothers, compromised the weaned foal’s social behaviour, in a very significant way. The foals with their mothers would, in theory, have a very strong social network (bonds), in contrast with the weaned foals, which will be lacking maternal protection, in an environment where the other group would have maternal protection. This situation could exacerbate the differences among groups, making it very good for the foals with their mother but exaggerating the challenges in the weaning group. This could have been corrected adding another control group with weaned foals kept with unrelated mares, removing the impact of social network (previous bonds) on the behaviour of the foals. The study needs better behavioural data, to characterize social interactions in the group in a more detailed way.

We fully understand and appreciate the reviewer’s insightful comment regarding the potential impact of group composition and social dynamics on our findings. We acknowledge that the presence of both weaned and non-weaned foals within the same social group likely shaped the social environment in a way that was not neutral. In fact, this was a key part of our rationale: the study aimed to assess whether peers and unrelated adults could buffer the effects of weaning in a shared social environment, without isolating individuals from their conspecifics. As the reviewer rightly points out, the environment was likely more socially secure for foals that remained with their mothers, and more socially challenging for the weaned individuals. This imbalance is not viewed as a flaw in the design but rather as an important feature of the ecological validity of our experimental setting. Unlike most previous studies where weaned individuals are fully socially isolated or housed between weaned animals only, here all foals remained in social groups, which allowed us to evaluate the limits of peer and adult presence in mitigating the effects of maternal separation. The results suggest that while social partners

are available, they are not equivalent to the maternal figure in providing social support. We agree that adding a control group of weaned foals placed with unrelated adult females would have been a valuable addition to the experimental design. Unfortunately, mainly for logistical reasons, such a configuration was not possible in the context of this study. We do, however, see this as a very relevant direction for future research and we now state this more clearly in the conclusion section of the article.

Regarding our collection of behavioural data, we believe that the current scan sampling protocol already provides a rich and reliable dataset, including information on individual behaviour, social interactions, and spatial proximities. As mentioned above, we have now clarified in the revised manuscript the extent of the observational parameters recorded and added new supplementary material to better illustrate the structure of the social network within the two groups. This includes social network graphs based on proximity data, which show that the weaned foals maintained social connections with their peers and were not isolated within the group. Importantly, we did not observe signs of strong aggression or systematic rejection from other group members, which would suggest a hostile or highly asymmetrical environment. We fully agree, nonetheless, that future studies would benefit from more detailed behavioural data using focal sampling or continuous recording to allow for a finer characterization of social interactions and affiliative dynamics.

In summary, while we acknowledge the limitations raised by the reviewer, we believe that our current dataset is still solid and informative, and that the social configuration, though imperfect, reflects a meaningful intermediate scenario between full maternal presence and social isolation. We hope that the addition of the supplementary social network data and the clarifications brought to the manuscript will reassure the reviewer about the validity of our interpretations.

We now include all these perspectives within the Discussion section of the manuscript and the text was modified as follows :

Revised text (Lines 382-390): *“Nevertheless, an alternative hypothesis could posit that the presence of the mother might have provided indirect benefits through pre-existing affiliative bonds within the group. These additional social resources may have further amplified the contrast with weaned individuals, increasing the level of social challenge for weaned horses compared to foals benefiting from the presence of their mothers. Future studies using focal sampling or continuous recording to capture overall social interactions will be necessary to unmask the affiliative dynamics and disentangle the respective contributions of maternal presence and group-level affiliative structure to better understand how prolonged maternal presence shapes developmental trajectories.”*

The MRI data needs to be better discussed in the areas involved in the contrast between the weaned and non-weaned groups, to offer an understandable explanation. Amygdala and its role in fear responses and empathy is not explored. The hippocampus, as the hub for memory, cognitive processes and emotions is also not discussed appropriately. The thalamus and its role in controlling pain, could be mentioned. The relationship between the hypothalamus and endocrine regulation, more than weight gain, is also relevant to the work.

In this revised version, we have improved our discussion and chosen to address the role of the amygdala in depth regarding its weight in our statistical model. However, we did not discuss in depth neither the role of the hippocampus and thalamus or the relationship between the

la science pour la vie, l'humain, la terre

UMR (* nom de l'UMR)

Adresse

Code postal et ville

Tél. : 00 00 00 00 00

Rejoignez-nous sur :

Site internet de l'UMR

hypothalamus and endocrine regulation beyond weight gain. We chose to omit this from the discussion, firstly because of their limited weight in our statistical model, and to avoid describing the complex relationships between all these structures in our experimental context, which could confuse the reader and detract from the main message of the paper.

Revised text (Lines 281-296): *“Beside the DNM, we also observed functional modification within amygdala associated with social behaviors in foals. Amygdala is a hub structure involved in social processing but also in a large variety of behavioural processes including fear- and pain-related processes^{36,37}. Through its reciprocal connection with the aCC, a highly conserve brain network across mammalian evolution³⁸, amygdala has been described to influence social decision-making behaviors in monkeys³⁹. Indeed, when a monkeys choose to deliver juice rewards to a conspecific (positive other-regarding preference) an enhanced coherence is observed between the spiking of amygdala neurons and beta oscillations of the aCC but also between the spiking of aCC neurons and gamma oscillations in amygdala while when monkeys choose to deliver juice reward for themselves rather than a conspecific (negative other-regarding preference) these coherence are vanished³⁹. This interareal oscillatory coordination between the aCC and amygdala suggest that this circuitry control social behaviors. As such a circuitry is highly sensitive to early life stressors⁴⁰, an impaired functioning of this pathway could promote deficit of social skills³⁹. Here, our model shows that animals belonging to the “maternal presence” group display higher functional activity within both the aCC and amygdala (left/right), are more social and less isolated demonstrating that affiliative behaviors should impact the development of the functional coordination between aCC and amygdala.”*

Does the work meet the expected standards in your field?

Yes, I do consider the methodology sound and reflect the acceptable standards in our field of research.

Is there enough detail provided in the methods for the work to be reproduced?

More details are needed in the behavioural data. Blood collection is mentioned to have occurred without restraint of the animals, it that correct?

We thank the reviewer for this comment. Regarding the blood sampling procedure: yes, it is correct that no physical restraint was applied. The foals were guided through a narrow corridor between two barriers as part of a routine procedure to step onto the scale. This setup naturally limited their movements without requiring any form of strong restraint. Blood samples were taken swiftly and calmly by experienced experimenters during this brief passage. We added these details in the Methods section:

Revised text (Lines 706-711): *“To facilitate blood collection and weighing, the foals were gently guided through a passage formed by a system of metal barriers leading to a scale. A small front gate was momentarily closed to stop the foal briefly on the scale. Blood samples were drawn swiftly during this short pause—lasting only a few seconds—by an experienced handler standing alongside the foal. The gate was immediately opened after sampling. This setup allowed us to safely position the animal without halters or physical immobilisation.”*

As for the behavioural data, as stated earlier, we provided additional material as Supplementary Material and revised the Methods section to make to improve clarity and reflect the extent of our behavioural analyses.

We hope these additions will help ensure full reproducibility of our study.

Conclusion

In summary, this is an outstanding manuscript which needs some improvements to be considered for publication, particularly related to the behavioural data collection. I would like also to read an alternative hypothesis mentioned in the discussion acknowledging the possibilities that the experimental design may have exacerbated the negative responses of the weaned foals.

We thank the reviewer for this important suggestion. We agree that the composition of the social group may have contributed to amplifying the differences observed between weaned and non-weaned foals. In the revised manuscript, we now acknowledge that, beyond the direct effects of maternal presence, foals kept with their mothers likely benefited from additional affiliative support through pre-existing social bonds within the group. In contrast, weaned foals lacked these structured connections, potentially making the social environment more challenging for them. This complementary interpretation has been explicitly added to the Discussion section, along with a perspective for future research aimed at disentangling the respective contributions of maternal presence and group-level affiliative structure.

la science pour la vie, l'humain, la terre

UMR (* nom de l'UMR)

Adresse

Code postal et ville

Tél. : 00 00 00 00 00

Rejoignez-nous sur :

Site internet de l'UMR

INRAE

Reviewer #2 (Remarks to the Author):

The study by Valençon et al. used MRI measurements (anatomical, functional, and diffusion), along with physiological and behavioral measurements, to investigate the effects of caregiver loss on brain parameters and individual behavior. The study is very well written and represents a milestone in the field of social neuroscience. A key strength of the study is that the authors have carefully considered all the dependent variables in terms of MRI, physiology, and behavior. By combining multimodal imaging, they have created an atlas that will be essential for future MRI studies of the hours, which could make this work highly influential. These strengths demonstrate the study's importance to the field and make it a valuable contribution to publish here.

However, despite these strengths, I have identified some issues that need clarification before the manuscript can be considered for publication. My primary concern is the timing of the MRI imaging and the absence of a baseline measurement prior to separation. Additionally, fMRI is an indirect measure of brain activity, which can be influenced by physiological changes, potentially leading to a misinterpretation of brain network changes as functional plasticity. The authors should revisit the results and carefully address this issue to avoid such misinterpretation.

I have listed my comments below and would be happy to see an improved version of the manuscript after revisions.

We would like to thank the reviewer for his/her kind words and for helping us to improve the quality of the manuscript. We have responded point by point to all the concerns raised by the reviewer.

Major Comments:

1. Baseline Measurements:

In the study, sampling and monitoring began one month after separation. This results in a missing baseline measurement. Why did you not record most of the measurements before separation to establish a baseline for comparison?

This is a totally legitimate question, which has been raised by all the authors during the writing of the current manuscript.

Regarding metabolites and hormones concentrations, we choose to miss this as we did not find any statistical differences (Mann-Whitney test, p value set < 0.05) for metabolites and hormones concentrations 1 month before maternal separation. We introduced both metabolites and hormones concentrations evaluated before weaning within the supplementary material (**Table S3**) and data have been compiled within a new supplementary figure (**Fig S10**).

The other aim of our study is to evaluate how social behaviors and particularly how affiliative behaviours impact the behavioural development of offsprings. In this way, we assessed activity budgets, spatial proximity, centrality and spontaneous behavioural activity of each animal after the weaning procedure. Here some social metrics such as centrality were impacted by weaning procedure. Indeed, sociograms before and after weaning do not have a similar number of nodes (representing each animal: mares + foals) and edges (numbers of possible interactions) therefore, weaning irremediably changed the global topography of the social network of all the animals included in the study and no rigorous comparisons are possible before and after maternal separation. Therefore, we choose to miss the behavioural measure before weaning procedure.

la science pour la vie, l'humain, la terre

UMR (* nom de l'UMR)

Adresse

Code postal et ville

Tél. : 00 00 00 00 00

Rejoignez-nous sur :

Site internet de l'UMR

INRAE

Finally, animals included within this study come from a breeding facility which ensure a long-term monitoring of animals from several generations (the herd has been established in 1970). The control of the genetical homogeneity and stability of the herd is ensured by the experimental unit which developed a herd management procedure allowing to provide standardized animals for their both physiology and behaviors whilst ensuring to regularly introduce Weshl-b male to limit consanguinity (common parent limit 4th/5th generation). Furthermore, before our experimentation, veterinarian consultations have been performed to validate the state of animals' health (physical auscultation, vaccination, etc.). Hence, like for studies in rodents' models, horses and foals included in our study were provided by a breeding facility whose the practices allow to study both physiology and behavior in a cohort of standardized animals with an additional level of quality as animals regularly meet veterinarians before experimentations. Additionally, all the animals have been raised in similar conditions of breeding with an egal access to food, water, housing, etc. and a minimal human intervention, noise or any other disturbance. Taken together, these gold standard breeding conditions, allow to work (at first approximation), with standardized animals reducing both biological noise and dispersions of the data and this conclusion was reinforced by the absence of significant variations of both metabolites and hormones as see previously.

As we agree that biological variations (especially in small samples studies like here) could bias our results we randomly assigned mother-offspring dyads to “maternal presence” and “maternal absence” treatment, and we balanced those groups according to date of birth, weight and paternal origins as an additional level of rigor to reduce the risk of bias between both groups. We previously indicated the point within the method section.

Please see the “*Assessment of endocrine and metabolic phycological markers*” section within the supplementary for further information.

2. Timing of MRI Measurements:

Why were MRI measurements conducted only one month after separation? If one month is sufficient to observe the effects of caregiver loss, why were other measures recorded at 3, 5, and 7 months?

Our initial plan was to scan several time animals nevertheless foals reached 2 months after the weaning the limit of weight allowed by scanner device (<300kg). Hence, due to technical limitation we didn't scanned animal at 3, 5 and 7 months.

3. Lack of Controls and Quality Assessments:

Since this study is a pioneering effort in the field, it lacks several necessary control measures and quality assessments.

- You used Spin-Echo EPI for resting-state fMRI. However, in both animal and human research, Gradient-Echo EPI (GE-EPI) is the standard sequence. Could you clarify why you chose Spin-Echo EPI instead? Please add a supplementary figure showing raw images for quality assessment.

Firstly, we should bring to reviewer knowledges, that we used (only) three living animals (not included to the final analysis) to test and optimize our sequences in foals. For each animal, veterinarians imposed a 3h maximum limit for anaesthesia duration to limit myositis occurrence and consequently, we acknowledge that our sequences have a significant room for improvement. At the beginning of the project, our goals were to get a minimal TR, good SNR and tSNR, minimal distortions and susceptibility/truncate artefacts the whole in a minimum of

la science pour la vie, l'humain, la terre

UMR (* nom de l'UMR)

Adresse

Code postal et ville

Tél. : 00 00 00 00 00

Rejoignez-nous sur :

Site internet de l'UMR

INRAE

time to get anatomical, diffusion and functional data in a time compatible with experimental constraints.

It's true that GRE is a commonly used sequence for fMRI in human research but is not a really a standard in preclinical research to date. In our previous paper with Grandjean et al 2023 (Nature Neuroscience) our world-wide consortium of preclinical imaging aggregated 65 functional imaging datasets acquired from rats across 46 centres and both SE and GE sequence were indistinctly used by preclinical imaging community at least for rat. It's a matter of debate but to date the preclinical imaging community did not specifically recommend GE or SE for functional imaging, both having pros and cons.

If I acknowledge that GE sequences are definitively less sensitive to motion, faster and flexible, SE-EPI are better to capture BOLD compared to GE-EPI due to its sensitivity to microvasculature rather than macrovasculature (T_2). But the main reason for using a SE sequence in horse is the sensibility of GE (at least on our 3T siemens Verio Scanner 10 years old) for susceptibility artifacts. Indeed, SE sequences are particularly beneficial in brain regions near air-tissue or bone-tissue interfaces, where geometric distortions are problematic and regarding the size of frontal sinus, guttural pouch and eustachian tubes, and, by experience, these kinds of artefacts could be dramatic for the final functional images. With our sequences we preserved BOLD signal within frontal areas, amygdala, hypothalamus, etc. with a good SNR and tSNR.

We added two figures within the supplementary data (**Fig S3** and **Fig S4**) to allow readers to appreciate the quality of the raw data collected in our work and to visualize the constrain due to large cavities for fMRI acquisitions in this specie.

- It is essential to examine FA maps in subject space before running comparisons. Please provide these maps along with the FA value range.

Here, I don't understand the exact meaning of the reviewer's request. Do you would like me to provide the FA maps for the 23 animals? As previously mentioned, all MRI data could be downloaded from the EU Open Research Repository ZENODO and each FA maps could be easily calculated using the method available within the provided github or with FSL or other specialized software. Nevertheless, the reviewer's request is justified as this is a pioneering study. We added in the supplementary data (**Fig S9**) a representative FA map and RGB fiber orientation maps calculated from raw data (**Fig S9a**) and group level FA and RGB maps (**Fig S9b**) calculated from spatially normalized data. With this figure I hope to properly address the reviewer's wishes.

- For all MRI data (DWI and fMRI), please report SNR (Signal-to-Noise Ratio) and tSNR (temporal SNR) where applicable.

All SNR and tSNR have been calculated for each sequence and for each animal and have been compiled within the **Table S2** in the supplementary data file.

- It is crucial to report motion parameters for resting-state fMRI, as motion can easily introduce artificial correlations.

Both translation and rotations during fMRI acquisition have been calculated and reported in the supplementary data document within the **Fig S5**. Translations do not exceeded 1mm and rotations 0.3° .

- For resting-state fMRI, you used a ~4-second repetition time (TR), which is quite long. Such a low sampling rate may lead to aliasing effects, where respiration-related fluctuations modulate resting-state fMRI signals. Since caregiver loss affects physiological parameters, the observed changes in resting-state networks could be driven by physiological variations rather than genuine functional plasticity of brain regions. This needs a clear explanation. If respiration data were recorded, it would be valuable to report the respiration rate for both groups during MRI scans. Additionally, as shown in this study (<https://pmc.ncbi.nlm.nih.gov/>), respiration can introduce artifacts in rs-fMRI and is also associated with a respiration-related brain network. This factor should be considered when interpreting the results.

1/ Reviewer is right and it was our wish to decrease at the minimum the TR. The main issue here is that we've a field of view close to humans (22cm x 22cm x 13.2cm) but we needed to add large saturation bands on guttural pouch and frontal sinus to limit folding artefact during acquisitions and to limit distortions in both DWI and fMRI images. So, all this specific adjustment involves a TR close (but less) than 4s in our 3T siemens with a 45mT/m gradient. We acknowledge that the recording of both respiratory and cardiac rates would have a major impact to control the aliasing effect due to this long TR but, our 3T-compatible recording devices were not able to record anything probably due to the large chest of the animals and/or because of anatomical characteristics of foals mismatched with the human anatomy for who those devices are designed. Although, veterinarians did not observe significant modifications of breath rate or cardiac pulse between imaging sessions, their measure was a clinical assessment and do not constitute a true measure as both rates were investigated in a discrete manner instead of longitudinally. So, to reduce the temporal aliasing effect which could occur with such a TR, we applied a slice-timing correction (see the functional imaging data analysis section within the Material and Method in the main document) to adjust differences in acquisition times between slices to reduce temporal misalignment. We also remove high-frequency components from the global signal to prevent aliasing artefacts. Regarding networks obtained using the dictionary learning approach and those obtained using the seed-based analysis of the DMN, we suggest that we reached to control the aliasing effect due to our quite long TR.

As, the control of aliasing effects is not optimal we choose to perform an fALFF analysis to study the functional modification induced by maternal separation. Indeed, fALFF investigate the fluctuations of spontaneous low-frequency (0.01–0.08 Hz) and measures the relative contribution of low frequency fluctuations within this specific frequency band to the whole detectable frequency range (Zou et al., 2008). Low frequency fluctuations (LFFs) allow to study the amplitude of regional neuronal activity for identifying brain areas with significant modifications of local functioning between two conditions (Chen et al., 2015). Finally, by combining electroneurophysiological recordings and fMRI, many studies have suggested that the LFFs of blood oxygena-level-dependent (BOLD) fMRI signals are closely related to the spontaneous neuronal activities (Goldman et al., 2002; Logothetis et al., 2001; Lu et al., 2007; Mantini et al., 2007). Altogether, our methodological constrains and previous literature report that fALFF is the best metric to explore the functional modifications occurring within foal's brain. In conclusion, we do our best to limit temporal aliasing artefacts and we propose some paths within the discussion section of the main document to address this point.

Revised text (Lines 353-359): *“Finally, our pioneering study offers to the community a large range of acquisitions protocol for in vivo MRI scanning in foals (see supplementary methods for details). These protocols delivered here, could be considered as a good start for research/veterinary centres interested by brain imaging of equidae. Nevertheless, our*

la science pour la vie, l'humain, la terre

UMR (* nom de l'UMR)

Adresse

Code postal et ville

Tél. : 00 00 00 00 00

Rejoignez-nous sur :

Site internet de l'UMR

INRAE

sequences have a significant room for improvement (TR of functional data, resolution of diffusion data, contrast of anatomical data) and a multisite-based strategy will be necessary to stress our protocols and improve their quality.”

2/ The reviewer is right that as caregiver loss affects physiological parameters, the observed changes in resting-state networks could be driven by physiological variations rather than genuine functional plasticity of brain regions. Indeed, our statistical model show which variables are correlated between them and at this step we cannot conclude if the brain functioning is responsible for physiological modifications or if these latter modified brain functioning. Investigating such a question would require additional studies in which, for example, the DNM network would be manipulated (increased or decreased) by genetical or pharmacological approaches. Unfortunately, to date, the horse model is not adapted to address such a question using such ways but aa more controlled study with additional MRI sessions, longitudinal recording of behaviors and more frequent physiological assessments with more data point could allow to investigate this question.

This has been addressed within the conclusion section of the main document.

Revised text (Lines 390-396): *“Future investigations will also have to determine whether the changes detected in brain functioning caused by maternal loss are responsible for the physiological changes observed in foals or if it is the physiological modifications that lead to altered brain functioning. To date, both hypotheses remain open. Multiple imaging sessions, long-term behavioural recording, and closer physiological monitoring will be necessary to clarify which changes (brain, behavior, or physiology) occur first to find the directionality of the correlations detected here.”*

4. Statistical Tests and Reporting Issues:

- The statistical tests used in the study are unclear. You frequently mention Student’s t-test, but since you are comparing two groups, paired t-tests would be more appropriate. Were these tests one-tailed or two-tailed? This needs to be explicitly stated in the text.

Two-sample t-test is used when the data of two samples are statistically independent, while the paired t-test is used when data is in the form of matched pairs. As we cannot perform paired t-tests as we cannot constitute matched pairs, we performed two-sample t-test mentioned as Student T-test within the main document. We changed “Student T-test” by “Two-sample t-test”.

Secondly, this is a two tailed test, but the t-values reported at $p < 0.05$ for a dof =21 is 1.7207 when comparison is unweaned > weaned and 1.7207 when comparison is unweaned < weaned. So here t-value is a threshold to highlight significant voxels.

The t-value reported here is calculated by SPM for a given p-value and it used as a threshold to reveal significant clusters.

Here an example with Fractional Anisotropy results (**Fig 4a** of main document):

Unweaned < Weaned

Here the threshold used is 1000 voxels to reach FDR corrected

- You applied FDR correction for multiple comparisons, but FDR only provides a corrected p-value. What cluster size threshold was used to define significant clusters? This information is missing.

The cluster size threshold depend to test considered. The reviewer will find below the statistical summary provided by SPM for FA analysis. Here, the cluster threshold to reach FDR-corrected significant cluster is 15000 vx.

Statistics: p-values adjusted for search volume

set-level		cluster-level				peak-level					mm mm mm		
p	c	$p_{FWE-corr}$	$q_{FDR-corr}$	k_E	p_{uncorr}	$p_{FWE-corr}$	$q_{FDR-corr}$	T	(Z_E)	p_{uncorr}			
0.981	3	0.013	0.017	54275	0.000	0.606	0.989	4.45	3.69	0.000	-2.8	-48.6	8
						0.622	0.989	4.42	3.68	0.000	-7.6	-38.2	0
						0.847	0.989	4.05	3.44	0.000	-7.6	-43.8	28
		0.828	0.994	13070	0.046	0.985	0.989	3.58	3.13	0.001	-24.4	-36.6	-19
						1.000	0.989	3.05	2.74	0.003	-33.2	-39.8	-14
						1.000	0.989	2.90	2.63	0.004	-46.8	-18.2	-6
		0.957	0.994	9569	0.082	1.000	0.989	3.05	2.74	0.003	-0.4	32.2	-6
						1.000	0.989	2.57	2.37	0.009	0.4	28.2	20
						1.000	0.989	2.51	2.33	0.010	4.4	21.0	24

For VBM comparison a cluster threshold of 6000vx is sufficient.

Statistics: p-values adjusted for search volume

set-level		cluster-level				peak-level					mm mm mm		
p	c	$p_{FWE-corr}$	$q_{FDR-corr}$	k_E	p_{uncorr}	$p_{FWE-corr}$	$q_{FDR-corr}$	T	(Z_E)	p_{uncorr}			
0.073	2	0.000	0.000	24762	0.000	1.000	1.000	3.91	3.35	0.000	38.0	-36.6	-10.
						1.000	1.000	3.59	3.14	0.001	-26.0	-11.8	1.
						1.000	1.000	3.42	3.01	0.001	18.8	-45.4	-12.
		0.272	0.199	5419	0.002	1.000	1.000	3.48	3.05	0.001	-34.0	-11.8	-3.
						1.000	1.000	3.24	2.88	0.002	-34.8	-3.8	-8.
						1.000	1.000	2.88	2.62	0.004	-34.8	9.0	-5.

For DNM comparison a cluster threshold of 500vx is sufficient.

Statistics: p-values adjusted for search volume

set-level		cluster-level				peak-level					mm mm mm		
p	c	$P_{\text{FWE-corr}}$	$q_{\text{FDR-corr}}$	k_E	P_{uncorr}	$P_{\text{FWE-corr}}$	$q_{\text{FDR-corr}}$	T	(Z_E)	P_{uncorr}			
0.000	4	0.000	0.000	641	0.000	0.000	0.000	10.91	6.07	0.000	-43	-44	13
		0.000	0.000	966	0.000	0.000	0.000	10.22	5.90	0.000	41	-38	11
						0.920	0.522	4.18	3.48	0.000	54	-28	3
		0.000	0.000	1970	0.000	0.001	0.001	9.40	5.67	0.000	-3	21	-8
						0.003	0.003	8.32	5.34	0.000	-7	24	2
						0.026	0.019	6.87	4.81	0.000	-5	28	-14
		0.000	0.000	1347	0.000	0.073	0.036	6.23	4.54	0.000	6	-43	8
						0.104	0.039	6.01	4.44	0.000	-5	-41	13
						0.135	0.047	5.84	4.37	0.000	1	-40	18

All these information's have been added within figures legends.

- In Figure 3, I see small blue clusters (e.g., first image, front; middle image, top row) and yellow clusters (present in all figures except the first top left). However, these are not mentioned in the results. Please clarify and update the text accordingly.

All the clusters are not described because some of them do not survive to FDR correction. This is the case for these clusters.

- Another surprising issue is that many reported t-test values are identical ($t = 1.7207$) for different comparisons (line 126 for GMC, line 141 for FA, and line 144 for ODI). This seems highly unlikely. Please double-check your test results.

Because the t-value used here (1.7207) is not the t value of the test but the value used to threshold the t-maps. All the voxels with a t-values upper the threshold are considered as significant.

- The value $t = 1.7207$ is quite low and corresponds to the critical value of a one-tailed t-test. Please clarify whether this was intentional.

In SPM convention is the tvalue used for one tail. As it's a two-tail t test the value is $1.7207 \times 2 = 3.4414$. Considering this we replaced "t = 1.7207" by "t= 3.4414, two-tailed" within the main document.

- Wherever possible, report the actual p-values instead of just stating significance thresholds.

Added information should help readers to understand statistics.

5. Issues in Figure 4 Reporting:

- In Figure 4a, a yellow cluster suggests a significant effect during maternal presence, but this is not reported in the results. Why?

All the clusters are not described because some of them do not survive to FDR correction. This is the case for these clusters.

- Similarly, in Figure 4b, a blue cluster is visible but not discussed. These findings should be addressed in the text.

All the clusters are not described because some of them do not survive to FDR correction. This is the case for these clusters.

6. Default Mode Network Identification:

- How did you define the default mode network (DMN)?

The default mode network (DMN) of equidae has been identified from its three canonical core clusters, the medial frontal cortex, precuneus region and posterior cingulate/retrosplenial cortices, and angular cortices as reported by litterature¹⁵⁻¹⁷.

la science pour la vie, l'humain, la terre

UMR (* nom de l'UMR)

Adresse

Code postal et ville

Tél.: 00 00 00 00 00

Rejoignez-nous sur :

Site internet de l'UMR

- What was the correlation value between these regions?

A summary of the mean correlation between each ROI of the DMN were included in the supplementary material document. Mean correlation values span from 0.35 (angular cortex left/anterior cingular cortex) to 0.82 (angular cortex right/retrosplenic cortex). See Figure S9C within the supplementary.

- Were there additional regions correlated with this network that were not reported?

No.

- Why was $p < 0.001$ used as a significance threshold instead of $p < 0.05$?

For clarity of the figure, the significance threshold has been changed from 0.05 to 0.001. Please see the “*default mode network of equidae*” section within the supplementary to see the difference between these two thresholds.

7. Multiple Regression Analysis and Default Mode Network (DMN) Interpretation:

- The multiple regression analysis used to identify the DMN is not well explained. Please provide a detailed description of the methodology.

For details, please see the “*Large scale networks identification*” and “*The default mode network of equidae*.” section within the supplementary information document.

- How did you detect anticorrelation between somatosensory regions? More details are needed to support this claim.

For details, please see the “*Large scale networks identification*” and “*The default mode network of equidae*.” section within the supplementary information document.

- Was this network also visible in ICA (Independent Component Analysis)? If so, please clarify.

Yes, please see the “*Large scale networks identification*” section within the supplementary information document for details.

8. Figure 8 and MRI Metrics:

- What MRI-derived metrics were used to plot the matrix in Figure 8?

76 variables have been collected for the analysis: 18 from MRI imaging, 34 from behavioural follow up, and 24 physiological follow up. All the variables used are summarized within the supplementary document “variables.csv”.

- Since MRI was recorded only one month after separation, but many behavioral and physiological measures were recorded much later (e.g., behavioral tests at 7 months), there is a significant time gap. Given the aging effects on MRI-derived parameters, comparing these results with physiological or behavioral measures recorded much later is problematic. This limitation needs to be discussed in detail.

We now address this question by the previous addition did within the conclusion section of the main document.

Revised text (Lines 390-396): “*Future investigations will also have to determine whether the changes detected in brain functioning caused by maternal loss are responsible for the physiological changes observed in foals or if it is the physiological modifications that lead to*

la science pour la vie, l'humain, la terre

UMR (* nom de l'UMR)

Adresse

Code postal et ville

Tél. : 00 00 00 00 00

Rejoignez-nous sur :

Site internet de l'UMR

INRAE

altered brain functioning. To date, both hypotheses remain open. Multiple imaging sessions, long-term behavioural recording, and closer physiological monitoring will be necessary to clarify which changes (brain, behavior, or physiology) occur first to find the directionality of the correlations detected here."

9. Misinterpretation of fMRI Findings in the Discussion:

- The statement in the discussion: "Providing a better understanding of how structural and functional brain organizations are linked with physiology and/or behavior" is misleading.

Within reference 33 of the main manuscript authors wrote:

Introduction: "multivariate methods are better suited to capture brain-behaviour relationships." and "the two most used types of PLS in neuroimaging are Partial Least Squares Correlation (PLSC) and Partial Least Squares Regression (PLSR). PLSC is a correlational technique that estimates associations between two sets of data (e.g., behaviour and brain morphology), while PLSR is a regression technique that predicts one set of data from another (e.g., predicts behaviour from brain activity)."

Here we used a type of Partial Least Squares Correlation (PLSC) and as mentioned By Viera et al 2024, the aim of such analysis is to map the relationship between brain metrics and biobehavioral datas.

Here we don't understand the reviewer point and we maintained our sentence.

- fMRI measures BOLD signals, which depend on blood oxygenation levels. Physiological changes can easily influence these signals, meaning that observed correlations between physiology and BOLD signals should not be directly interpreted as functional plasticity. The results do not necessarily indicate brain function is changing but rather that physiological variations may be driving the observed effects.

- A more physiology-independent technique is needed to validate these findings.

- Additionally, the papers cited in support of this claim focus on disease symptoms and cognition, not physiological changes and brain networks. Please **clarify this distinction**.

This three question seems redundant and have been addressed by the previous addition did within the discussion section of the main document.

Revised text (Lines 390-396): *"Future investigations will also have to determine whether the changes detected in brain functioning caused by maternal loss are responsible for the physiological changes observed in foals or if it is the physiological modifications that lead to altered brain functioning. To date, both hypotheses remain open. Multiple imaging sessions, long-term behavioural recording, and closer physiological monitoring will be necessary to clarify which changes (brain, behavior, or physiology) occur first to find the directionality of the correlations detected here."*

10. Discussion Needs Revision Regarding Physiological Effects on BOLD fMRI:

- The discussion needs to acknowledge that BOLD signal variations may be due to physiological changes rather than true functional brain network changes due to the mother separation.

- Please refer to the following studies, which discuss the impact of physiology on BOLD signals: <https://www.sciencedirect.com/> and <https://www.mdpi.com/2072->

la science pour la vie, l'humain, la terre

UMR (* nom de l'UMR)

Adresse

Code postal et ville

Tél. : 00 00 00 00 00

Rejoignez-nous sur :

Site internet de l'UMR

- The interpretation of fMRI results should be revised accordingly.

Here the reviewer is back to his/her previous questions (3-5 and 9-2,9-3,9-4). As previously mentioned, we first did our best to control the temporal aliasing effect and the additional information provided within the supplementary seems good enough to demonstrate that we limited this effect. Secondly, we addressed the hypothesis of the role of physiological modifications within the observed brain modifications in the conclusions section.

Finally, we think that the literature provided in this article is accurate and sufficient to allow reader to understand the two previous points.

Minor Comments

1- Figure Quality: The quality of the figures is too low, making the numbers and labels unreadable. Please improve the resolution to ensure readability.

All the figures are proposed on svg format to provide the better resolution.

2- Figure 1g: The label for Figure 1g is missing. Please ensure it is clearly labeled on the figure.

This has been modified see new figure 1.

3- Figure 1d Clarification: In Figure 1d, it is unclear whether the test was done only once after 7 months of maternal separation, or if it was repeated at 1 month, 3 months, and 5 months as well. Please clarify and include this information in Figure 1a.

This has been clarified see new figure 1.

4- Line 100 - "Post-treatment": In line 100, the term "post-treatment" is used. Could you clarify what this refers to?

Here, we replaced "Post-treatment" by "post-weaning" for clarity.

5- Line 105 - Summary of Variables: In line 105, you mention "a total of 76 variables." Could you provide a summary of the variable names here to clarify for readers which variables were included?

An additional file named "*variables.csv*" has been provided to allow readers to understand the metrics used in the study.

6- Multiple Factor Analysis (MFA): What was included in the MRI imaging for the Multiple Factor Analysis (MFA)? The other variables are quantified, but how were these MRI variables quantified? Please clarify this process. It would be helpful to add a supplementary table listing all the variables used for this analysis for better clarity.

An additional file named "*variables.csv*" has been provided to allow readers to understand the metrics used in the study.

7- Figure 1 – Dim 1 and Dim 2: In Figure 1, could you explain what Dim 1 and Dim 2 represent?

As for PCA, each dimension of the MFA is a linear combination of the original variables, and the coefficients (loadings) indicate the contribution of each variable to the dimension. These dimensions capture the shared variability among the different groups of variables. They help identify the main trends or factors that are common across all groups.

8- Figure 1g Clarity: The labeling in Figure 1g is unclear. What exactly is plotted here? What do the light and dark colors represent? Why do some colors appear before time point 2? What

do the numbers on the horizontal axis mean? The figure suggests that MRI imaging is the least significant variable for separating the two groups. Could you discuss how the results would change if MRI measurements were excluded from the analysis?

The labeling in Figure 1g is unclear :

(f) using an integrative Multiple Factor Analysis where (f, left) shows the distribution of individuals along the first two axes of the MFA, and (f, right) displays the variables that contribute the most to the first MFA axis, which discriminates between the "with mother" and "without mother" animals.

What exactly is plotted here? What do the light and dark colors represent? What do the numbers on the horizontal axis mean?

The bars represent the contributions to the first axis of the MFA for each variable (light colors). The darkest bar corresponds to their coordinates on the same axis. A variable with a negative coordinate will be higher in the "without mother" group, which is also negative on the first dimension (f-left), and vice versa for positive coordinates. Therefore, the horizontal numbers correspond to the contribution or coordinates depending on the opacity of the bar being viewed.

Why do some colors appear before time point 2?

Here we didn't understand what reviewer means.

Could you discuss how the results would change if MRI measurements were excluded from the analysis?

As noted, although the imaging variables are discriminative, they are less so than the others, which alone are sufficient to separate the two groups.

9- Figure 1g – Text Readability: The text in Figure 1g is not readable. Please adjust the font size or improve the clarity of the text.

The font size has been increased to improve readability.

10- Line 115 - "Specific Brain Metrics": In line 115, you refer to "specific brain metrics." Could you clarify which metrics are included here? Please provide more detail about the brain metrics used in the analysis.

An additional file named "*variables.csv*" has been provided to allow readers to understand the metrics used in the study.

11- Figure 3 – Maximum Values: In Figure 3, what are the maximum values reported? Please add color bars with the corresponding maximum values visible in the figure to improve clarity.

Figure changed accordingly.

12- Line 157 – Frequency Band Selection: Why did you select the frequency band of 0.01–0.08 Hz? Please explain the reasoning behind this choice.

By focusing on the 0.01 to 0.08 Hz range, we minimized the impact of higher-frequency noise, (cardiac and respiratory cycles), which typically occur at frequencies higher than 0.1 Hz. Typically a horse breath 16 times by minutes and our visual inspection found that anesthetized horses breathed around 10 times per minutes (0.1 Hz). This range allowed us to study low frequency fluctuations.

13- Figures 6 and 7 – AUC Clarification: In Figures 6 and 7, could you clarify what is shown in the left plot of all subplots? Additionally, could you explain what AUC stands for and how it is being interpreted in this context?

The Area Under the Curve (AUC) measures capture the total effect over time, integrating both the magnitude and duration of responses. Unlike mean values, which provide a snapshot, AUC reflects the cumulative impact, revealing sustained effects and subtle differences that mean values might miss. This offers a more global and comprehensive view of behavioural and physiological changes.

la science pour la vie, l'humain, la terre

UMR (* nom de l'UMR)

Adresse

Code postal et ville

Tél. : 00 00 00 00 00

Rejoignez-nous sur :

Site internet de l'UMR

INRAE

Reviewer #3 (Remarks to the Author):

Review of “Affiliative behaviours regulate allostasis development and shape biobehavioural trajectories in horse”. This is an interesting study evaluating the impact of mother-separation in 6-month-old colts on short-term (6 month) measures of social behavior, physiology, and reactivity in a variety of behavioral novelty tests. The authors show that maternal removal from the herd (and hence maternal separation) also alters grey matter concentrations in areas relevant for emotional processing (e.g., amygdala, hippocampus, and hypothalamus) and the strength of Default Mode Networks, relative to mother-present colts.

Overall the results are explained appropriately. However, the links between maternal separation and the outcome measures remain correlational in nature, and language in the title and the discussion of the results should replace causal language (“affiliative behaviors regulate ... and shape biobehavioral trajectories” is too causal). The analyses show that the variables are correlated, but not necessarily causally related. As a consequence of this, the Discussion should also include a more detailed evaluation of alternative hypotheses that could account for the correlational differences between the two groups.

We agreed with the reviewer as indeed our PLS analysis seek correlation between collected variables. We addressed this within the discussion section of our study to fit with the point raised by the reviewer:

Revised text (Lines 390-396): *“Future investigations will also have to determine whether the changes detected in brain functioning caused by maternal loss are responsible for the physiological changes observed in foals or if it is the physiological modifications that lead to altered brain functioning. To date, both hypotheses remain open. Multiple imaging sessions, long-term behavioural recording, and closer physiological monitoring will be necessary to clarify which changes (brain, behavior, or physiology) occur first to find the directionality of the correlations detected here.”*

The data for the physiological measures (e.g., cortisol, cholesterol, triglycerides) reveal some interesting differences between the treatment groups. However, the authors do not report whether the two treatment groups were statistically equivalent in these measures prior to maternal separation. The Methods section indicates that a blood sample was collected just prior to maternal separation, so the samples are apparently available to analyze those data, but the paper only reports these measures (and the behavioral measures, as well) for the sampling periods AFTER maternal separation.

We added those data within the newest supplementary document (please see *Supplementary Results – Physiological Assessment*) and addressed the point raised by the reviewer.

The Introduction establishes ‘allostasis’ as an important organizing construct for this experiment, but the Discussion does not provide a sufficiently thoughtful discussion of how the data in the present experiment advise the concept that allostatic responsiveness is affected by maternal separation.

This point is critical and acknowledge that we did not provide a sufficient discussion in the previous version of the document. We correct this in this new version of the article:

Revised text (Lines 364-382): *“Despite being in a socially enriched environment with age-matched peers and adult conspecifics, the neurological, behavioural, and physiological development of foals is disadvantaged when they are coping with the absence of their mothers.”*

la science pour la vie, l'humain, la terre

UMR (* nom de l'UMR)

Adresse

Code postal et ville

Tél. : 00 00 00 00 00

Rejoignez-nous sur :

Site internet de l'UMR

This suggests the presence of a developmental control system nested at the social level in horses, a theory previously evoked for rodents, monkeys, and humans^{1,2,4,20}. For humans, Atzil et al. proposed a theoretical framework in which the authors hypothesize that, through social cues (care), parents foster the development of the allostatic regulatory system via the maturation of whole-brain neural networks, allowing for balanced biobehavioural development until adulthood. Basically, the allostatic regulatory system is responsible for the maintenance of physiological and behavioural states by making predictive and anticipatory regulations. To be predictive, this system must be fuelled by rich social interactions for the building of conceptual models from the detection of statistical regularities in interoceptive and exteroceptive inputs. The acquisition of these conceptual models is associated with the maturation of domain-general core networks in which interoceptive and exteroceptive inputs are constantly integrated to build models. This theoretical framework is consistent with our observations, as the absence of the mother reduced the size and functionality of the default mode network (DMN) in offspring in association with decreased sociality and impaired feeding behaviors and lipid metabolism. Our observations suggest that the theoretical framework of Atzil et al. for humans could be extended to horses and perhaps to numerous social mammals.”

The timeline for assessing the impact of mother-removal is very limited, given the longevity of horses. There are no data on the long term impact of this manipulation that would be necessary to conclude that maternal separation had an impact on long-term behavioral/neural/physiological) trajectories, beyond a short-term impact during the 6 months separation period studied in this paper.

That is a good point raised by the reviewer but it's another project. All the females are still housed within our facilities for herd management. Males have been sold at three years old. Consequently, long lasting measurement could be investigated in the future to see how early maternal separation impacted the biobehavioural development on long term but in two different studies focusing either on female or on male. For male, as a tight relationship exist between our research facilities and actual owners we could imagine send questionnaires to owners and/or plan visits to evaluate how early or late separation impacted adaptation to new environment, pairs relationship, weight course, etc. For females both behavioural and physiological data could be recorded in the future to investigate their biobehavioural trajectories in a more controlled environment.

Regarding imaging the large weight of animals restrain MRI investigation as weight limit of our scanner is set at 300kg. If some adjustments are feasible to allow MRI acquisition in adult, drug used, doses and duration of anaesthesia should be revisited to limit myositis occurrence.

Minor points/queries:

- The paper does not describe (or really even discuss) the significance of the Area Under the Curve measures reported for Behavior and Physiological measures. What additional information is made available by these measures above and beyond what is reported for mean values (rates of behavior or concentrations of analytes)?

The Area Under the Curve (AUC) measures capture the total effect over time, integrating both the magnitude and duration of responses. Unlike mean values, which provide a snapshot, AUC reflects the cumulative impact, revealing sustained effects and subtle differences that mean values might miss. This offers a more global and comprehensive view of behavioral and physiological changes.

- As far as this reviewer can determine, this manuscript presents the first data on MRI and fMRI data from horses from this laboratory. The description of the Methods for these measures, and the analyses performed on the data, are appropriate but lengthen (and potentially detract from) the main argument regarding the impact of maternal separation on behavioral, neural, and physiological function.

It's true that imaging methods described in this article are precisely explained and could discourage readers from ethology or basic neuroscience by complexifying the main message which is the role of affiliative behaviors on biobehavioural development. This article is positioned between three worlds namely the imaging, ethology and neuroscience communities. We tried to balance the manuscript in order to each community find relevant scientific and methodological information for its field.

That was a hard work and the multiple points raised by reviewer 2 show how the imaging community needed additional and (critical) information on the method used to study the equidae brain using MRI. For this reason, we choose to address a lot of technical point raised by the reviewer 2 within the supplementary method to allow for ethologists and neuroscientists out of the neuroimaging field to find information relevant for their fields.

Nevertheless, imaging is central within this study as it's a pioneering MR imaging study in horse as mentioned by the present reviewer and the second one. We tried again to balance the manuscript, and we hope that actual modifications will satisfy the current reviewer. We are open to discussion to improve the quality of the manuscript in futures correspondence.

- The paper includes subjects of both sexes, and presumably included 'sex' as a factor in the analyses. However, the paper does not describe whether no sex differences were detected or whether some of the changes in measures as a consequence of maternal separation differed between males and females.

The reviewer is right, and this point has been discussed by the consortium. We didn't choose to speak about the sex differences even if we acknowledge that such an effect should exist. The main reason is the number of animals involved in the study. Basically, the data analysis plan was based to investigate the effect of the early weaning on physiological and behavioural development and so here the H0 hypothesis seek to find the significant differences between two sex-balanced groups (weaned *versus* unweaned) with 12 animals per groups. To investigate the role the H0 hypothesis should be changed as the number of animals per group since a factor is added (female weaned *versus* female unweaned *versus* male weaned *versus* male unweaned). To compare all these groups, the number of animals necessary to reach the statistical power necessary should be largely increased and we reached the technical limitation of our facilities (housing). So, we choose to limit our investigation to the effect of early maternal loose in offspring independently of sex.

- The measure of social proximity (≤ 10 m from a potential social partner) seems like a liberal measure of social proximity. Some indication of the size of the paddock may help clarify this point.

We thank the reviewer for this helpful comment. We fully agree that the ≤ 10 m threshold may appear generous as a measure of social proximity. To clarify: this threshold was only used to compute a simple index of spatial isolation (i.e. being >10 m from any conspecific), designed to capture spontaneous distancing tendencies. It was not used for social network analyses. For those, we relied on a stricter threshold of <1 m, which is consistent with previous literature in horses and more appropriate for identifying meaningful affiliative proximities.

la science pour la vie, l'humain, la terre

UMR (* nom de l'UMR)

Adresse

Code postal et ville

Tél. : 00 00 00 00 00

Rejoignez-nous sur :

Site internet de l'UMR

It is worth noting that the >10 m isolation score revealed a transient group difference at the +1-month time point only, and did not persist over time. This suggests that social withdrawal or distancing was not a defining feature of the weaned foals. As now illustrated in the Supplementary Figures, the social network graphs confirm that weaned individuals remained well integrated within the group, further supporting this interpretation.

We believe that all these details were already provided in the original manuscript, but to avoid any potential confusion for the reader, we have now revised the structure of the “Behavioural observations” within the material and methods section of the main document and present the social proximity and network metrics first, followed by the more synthetic spatial isolation score:

Revised text (L.611-619): *“In addition, a centrality score, or eigenvector, has been calculated based on spatial proximities with the closest neighbour⁶¹. Two individuals were considered close when the inter-individual distance was less than 1m (i.e. distance enabling the individuals to interact without having to move). The eigenvector was obtained using SOCPROG 2.961 using a two-entries proximity matrix. The eigenvector represents the total number of connections an individual has, including the connections between its partners. The higher this index is, the more central the individual is within its social group. Finally, an “isolation” score has been calculated based on the proportion of scans spent isolated (+10m away from another groupmate). The higher the time spent isolated, the less gregarious and the more socially independent the foal is. All these parameters have been calculated for each foal for each period of observation (+1 month, +3 months, and +7 months post-maternal separation).”*

la science pour la vie, l'humain, la terre

UMR (* nom de l'UMR)

Adresse

Code postal et ville

Tél. : 00 00 00 00 00

Rejoignez-nous sur :

Site internet de l'UMR

INRAE